# Goal commitment is supported by vmPFC through selective attention

Eleanor Holton [1] ✉, Jan Grohn [1,2], Harry Ward[3], Sanjay G. Manohar [1,2,4,6], Jill X. O'Reilly[1,2,6] & Nils Kolling [2,5,6]

When striking a balance between commitment to a goal and flexibility in the face of better options, people often demonstrate strong goal perseveration. Here, using functional MRI ($n$ = 30) and lesion patient ($n$ = 26) studies, we argue that the ventromedial prefrontal cortex (vmPFC) drives goal commitment linked to changes in goal-directed selective attention. Participants performed an incremental goal pursuit task involving sequential decisions between persisting with a goal versus abandoning progress for better alternative options. Individuals with stronger goal perseveration showed higher goal-directed attention in an interleaved attention task. Increasing goal-directed attention also affected abandonment decisions: while pursuing a goal, people lost their sensitivity to valuable alternative goals while remaining more sensitive to changes in the current goal. In a healthy population, individual differences in both commitment biases and goal-oriented attention were predicted by baseline goal-related activity in the vmPFC. Among lesion patients, vmPFC damage reduced goal commitment, leading to a performance benefit.

In natural environments, many goals, whether it be pursuing prey, cooking dinner or preparing an article for publication, are only obtained after persevering through a substantial period of unrewarded time and effort. In all these cases, optimal behaviour requires balancing commitment to the current goal against flexibility to abandon it if the goal is no longer worth pursuing relative to alternatives. Psychiatry and neuroscience have tended to focus on 'failures' of commitment during extended behaviours[1–3]. However, behavioural economics provides us with ample examples of people showing 'too much' commitment to a goal, particularly after investing time or money[4–7]. These 'sunk-cost' biases are not unique to humans but have also been found in rodents[8].

Why might animals show biases towards overpersisting with a goal? When behaviour is structured by sequential goals, constant re-evaluation can be both expensive and distracting. In consequence, it has been proposed that distinct phases of 'deliberation' (evaluation of available options) and 'implementation' (committing cognitive resources to achieving the chosen goal) might be present in both humans and non-human animals[8–13]. However, a picture involving entirely discrete decision phases fails to explain how animals remain flexible to goal abandonment when the situation requires it. A plausible mechanism would allow for the agent to preferentially allocate processing resources to goal completion while retaining the necessary flexibility.

A candidate mechanism for such flexible focus on a goal is 'selective attention', specifically towards information about the chosen goal. Attentional selection need not be all-or-nothing but can vary in strength as the need to exclude distractors varies[14], thus allowing for flexibility. In ecological scenarios, we are faced with different reasons for abandoning a goal: progress might be too gradual or might reverse; alternatively, other options might become substantially more attractive. These different forms of pressure give rise to different emotional responses: frustration (with the current goal) in the former cases[10] and temptation (by alternative goals) in the latter. If selective attention

[1]Department of Experimental Psychology, University of Oxford, Oxford, UK. [2]Wellcome Centre for Integrative Neuroimaging (WIN), University of Oxford, Oxford, UK. [3]Centre for Experimental Medicine and Rheumatology, Queen Mary University London (QMUL), London, UK. [4]Nuffield Department of Clinical Neurosciences, University of Oxford, Oxford, UK. [5]Stem Cell and Brain Research Institute U1208, Inserm, Université Claude Bernard Lyon 1, Bron, France. [6]These authors jointly supervised this work: Sanjay G. Manohar, Jill X O'Reilly, Nils Kolling. ✉e-mail: eleanor.holton@psy.ox.ac.uk

to the chosen goal increases over the course of goal pursuit, this leads to a rather specific prediction about the interaction of 'temptation' and 'frustration' with increasing proximity to the goal: namely, sensitivity to the value of alternative goals ('temptation') should decrease more than sensitivity to the value of the chosen goal ('frustration'). Our first aim was to test whether attention and decision making showed these markers of increasing attentional orientation towards the current goal over the course of goal pursuit. To test this, we orthogonally vary the value of the current goal and the value of alternative goals at the time of decision, as well as continuously measure goal-oriented attention outside the decision period.

Our second aim was to investigate how goal commitment is achieved on a neural level. The ventromedial prefrontal cortex (vmPFC) has previously been shown to flexibly represent choice values according to the agent's current goal[15–20], linked to the compression of task-irrelevant information[21]. In addition to this body of research implicating the vmPFC in task-specific cognitive maps, a separate line of research has identified a key role for baseline vmPFC activity in carrying contextual information which biases subsequent choices in line with a previous behavioural strategy[22–24]. While the vmPFC represents attributes relevant to the current goal across extended timescales[25], the anterior cingulate cortex (ACC) has been shown to represent information about 'alternative' goals and the value of shifting away from the current strategy[26–32].

Using a novel task in combination with (1) computational modelling of behaviour, (2) functional magnetic resonance imaging (fMRI) and (3) behavioural analysis of patients with brain lesions, we investigated how goal commitment develops during goal pursuit. In our sequential choice task, participants advanced incrementally towards completing a chosen goal in the face of alternative goal offers. Participants showed a universal 'goal commitment' bias towards persisting with their current goal, even in circumstances when they would greatly benefit from abandoning it. We were able to measure several markers of selective attention to the current goal. First, as predicted by the attentional account, we found that decision making reflected goal-directed attention: as participants approached goal completion, their decisions remained relatively more sensitive to the value of the current goal than to the value of alternatives. Second, using a separate spatial working-memory task, we found that even outside the decision period, stimuli related to the current goal were increasingly prioritized in attention.

Using fMRI, we found that across participants, the degree to which baseline vmPFC tracked progress with the current goal predicted both attentional and decision-based metrics of goal capture. To probe the causal role of this signal, we ran the same paradigm in an independent sample of patients with brain damage; indeed, damage to the same area of the vmPFC identified in the fMRI study predicts lower over-commitment to the current goal resulting in a performance benefit.

## Results

### Primary decision task

Participants performed a 'fishing net' task with the aim of filling as many nets with seafood as possible over the course of the study (Fig. 1). Participants accumulated seafood 'goods' over several trials and only gained a reward when the net was full. On each trial, participants chose between offers for three types of good (octopus, crab or fish), where the quantity available for each good was shown by a green bar. Once selected, the offered quantity would be immediately added to the net. Importantly, only a single type of good could be collected in the net at once. This meant that if participants chose a different type of good from the type currently in their net, they would forfeit all their previously accumulated goods ('abandonment choice'). Alternatively, participants could choose to continue with the current goal by choosing to collect the same good already in the net ('persistence choice'; see Fig. 1a for example).

While the quantities offered for each type of good drifted gradually from trial to trial (random Gaussian walk with low variance), sometimes the quantity would drastically change for a given good (10% chance of a large shift up or down in offered quantity, independent of each type of good; see Fig. 1b for example offer trajectories across a block). If the quantity associated with the current goal collapsed ('frustration') or if an alternative good became much more bountiful ('temptation'), participants often benefitted from abandoning their progress and switching to an alternative good (Fig. 1b).

### Spatial attention task

Participants performed the decision task first inside the fMRI scanner and then in a separate behavioural session outside the scanner. Outside the scanner, in addition to the main decision task, participants performed an interleaved spatial attention task before every trial, providing a separate measure of attentional prioritization of the current goal (Fig. 1c, left). Participants viewed the stimuli associated with the three goods flashed on the screen and were then prompted to report the three item locations with a mouse click (stimuli were probed in a random order). While the spatial attention task involved the same 'seafood' stimuli, participants were explicitly told that memory performance would not impact subsequent offers in the decision task (see Extended Data Fig. 1 for full illustration of the task presentation and scanner variants).

### People show greater goal commitment than an optimal model

Because of the need to commit to a good for many trials to realize the reward (delivered on the completion of a full net), a good decision is based not only on the current offer, but also on the quantity already in the net and projections of future offers (see Extended Data Fig. 2). To understand how participants made such projections, we constructed a series of models reflecting increasingly complex possible strategies (see Methods and Extended Data Fig. 3 for details of models including validation and fitting procedures). The participants' behaviour was best described by the most complex model we tested ('tree-search model'; Fig. 2a), which provides an approximation of the optimal choice. This model samples possible future trajectories for the option offers using the true generative procedure and selects the option which is predicted to fill the net fastest when averaging across the sampled trajectories (Monte Carlo sampling of offer trajectories).

While choice strategies were best described by the tree-search model rather than simpler heuristics, people tended to overpersist with their current goal beyond the model's predictions. Persistence biases were quantified as an individual's deviation away from the tree-search model, in terms of their reluctance to abandon the current goal beyond the model's abandonment predictions (Fig. 2b; persistence biases are significantly greater than zero (Wilcoxon $Z = 4.78$, $n = 30$, $P = 1.73 \times 10^{-6}$)). This metric of persistence bias had good test–retest reliability within participants across sessions (intraclass correlation coefficient = 0.76, $P = 0.002$, 95% confidence interval (CI) = (0.25, 1.0); Extended Data Fig. 4d). Compared with the optimal model, people were more reluctant to abandon their goal the more progress they made towards finishing (main effect of proportion of net completed on top of tree-search model switch value: $X^2(1, N = 30) = 5.27$, $P = 0.022$; illustrated by binning in Fig. 2c; see Extended Data Fig. 3 for additional information about goal progress, and Extended Data Fig. 5 for comparison with tree-search model behaviour).

### Commitment is linked to higher goal-oriented attention

We predicted that goal-oriented attention and decision-making biases would be related during goal pursuit. To measure goal-oriented attention, we investigated how attention was distributed between stimuli associated with the current and alternative goals in a decision-free spatial attention task interleaved between decisions. Since the spatial attention task was not possible to perform using a button box inside

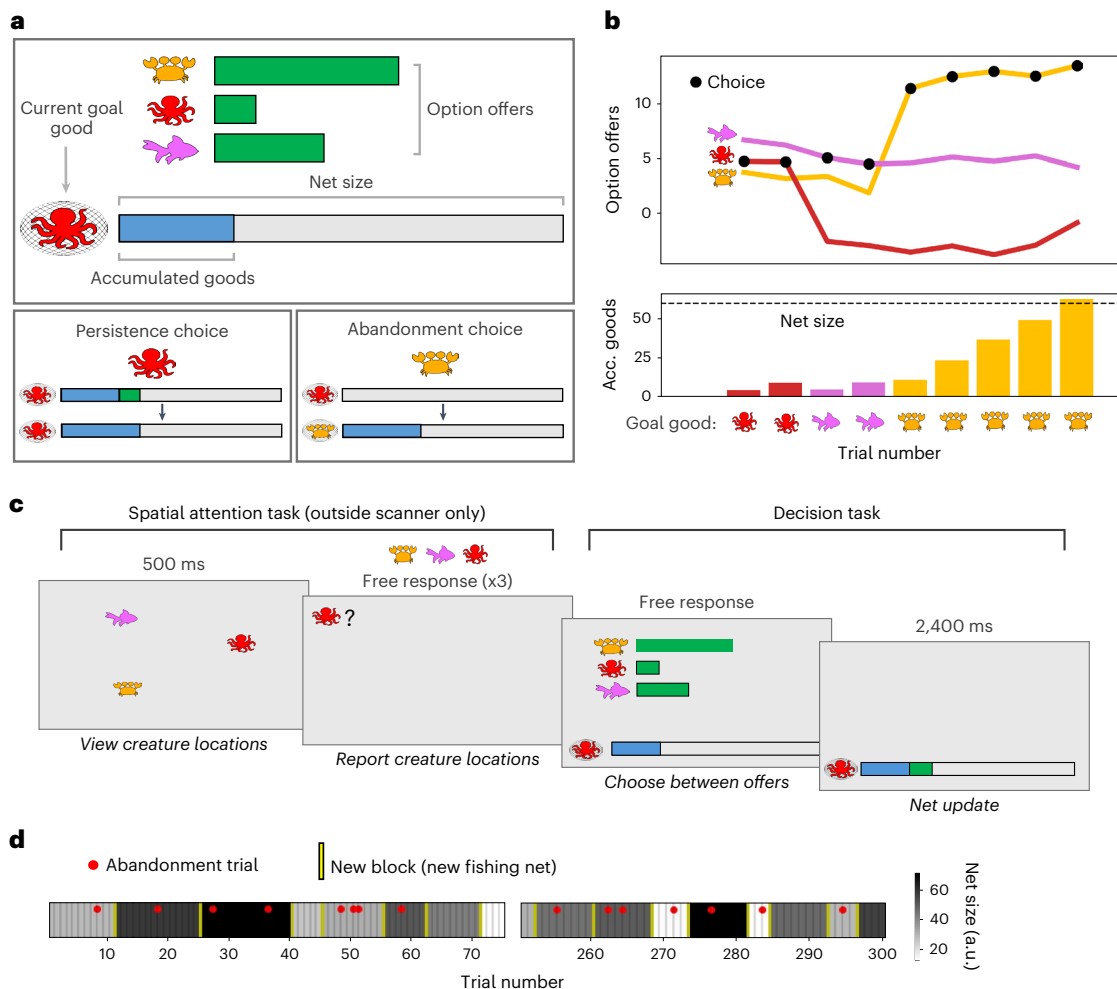

**Fig. 1 | Experimental design. a**, Participants performed a 'fishing net' task that involved incrementally filling nets with seafood 'goods'. Top: on each trial, bars indicate the current available quantities of each type of good (octopus, crab or fish) which participants could add to their net. The current net contents are shown in a separate bar at the bottom of the screen. Critically, since the net could contain only one type of good, switching goods meant forfeiting the pre-existing net contents. Bottom left: if participants continued with the same good, the offered quantity was added to the existing net contents. Bottom right: if participants chose a different good, the accumulated contents were emptied before the new goods were added. Participants received a single reward when a net was completed, and the net size and option offers were reset. **b**, An example block where a participant switches goods twice. Top: coloured lines depict the offers associated with each type of good across a block. Black dots depict the good chosen by the participant on each trial. During a block, the offers associated with each good varied gradually across trials with independent random walks, but could also jump to extreme high or low values (from where the random walk would continue). Bottom: bars depict the goods accumulating in the net, where icon and colour depict the type of good. Dashed line depicts the net size. **c**, Task sequence. Outside the scanner, participants performed the same decision task, with an additional interleaved spatial attention task performed on every trial. Participants viewed the three goods flashed on the screen in random locations and were then probed on the location of each good. Participants knew that their performance in the spatial task had no impact on subsequent offers. **d**, Example experimental timeline. The task always ended after a predetermined number of trials incentivizing participants to make strategic choices to maximize nets completed within the limited number of trials. Red dots indicate trials where the participant chose to switch. Shading indicates the varying sizes of the nets. Yellow lines indicate when a net was completed.

the scanner, we investigated these attentional biases in a separate testing session conducted outside the scanner. In the post-scan session, trials of the spatial attention task were interleaved with new trials of the main decision task.

In the spatial attention task, participants were asked to report the location of briefly flashed fish, octopus and crab symbols using a mouse click. Indeed, participants were both more accurate and faster at reporting the location of the currently pursued goal stimulus compared with the alternative goal stimuli (Fig. 2e; two-sided paired *t*-test; accuracy advantage for current goal: $t_{(29)} = 2.25$, $P = 0.032$, Cohen's $d = 0.42$; reaction time advantage for current goal: $t_{(29)} = 3.30$, $P = 0.003$, Cohen's $d = 0.61$). This accuracy difference was primarily driven by progressive memory enhancement for the goal stimulus: spatial accuracy for the current goal stimulus increased with the number of trials participants had been pursuing the current goal (Fig. 2f; effect

of pursuit time on goal item accuracy: two-sided *t*-test against zero, $t_{(29)} = -2.65$, $P = 0.013$, Cohen's $d = 0.44$; there was no significant effect of pursuit time on accuracy for alternative stimuli: two-sided *t*-test against zero, $t_{(29)} = -0.033$, $P = 0.974$, Cohen's $d = 0.006$). In a direct comparison, there was a significant difference between slopes for the effect of goal pursuit on selected and alternative goal items (Fig. 2f; two-sided paired *t*-test, $t_{(29)} = -2.37$, $P = 0.024$, Cohen's $d = 0.44$). This effect occurred even though the spatial task was performed outside the decision period and participants knew their performance on this interleaved task would not affect subsequent offers, suggesting a true attentional bias towards the chosen goal that increases with goal commitment.

This metric of attentional prioritization of the goal directly predicted individual differences in persistence biases: people who showed more goal-directed attention demonstrated higher persistence biases

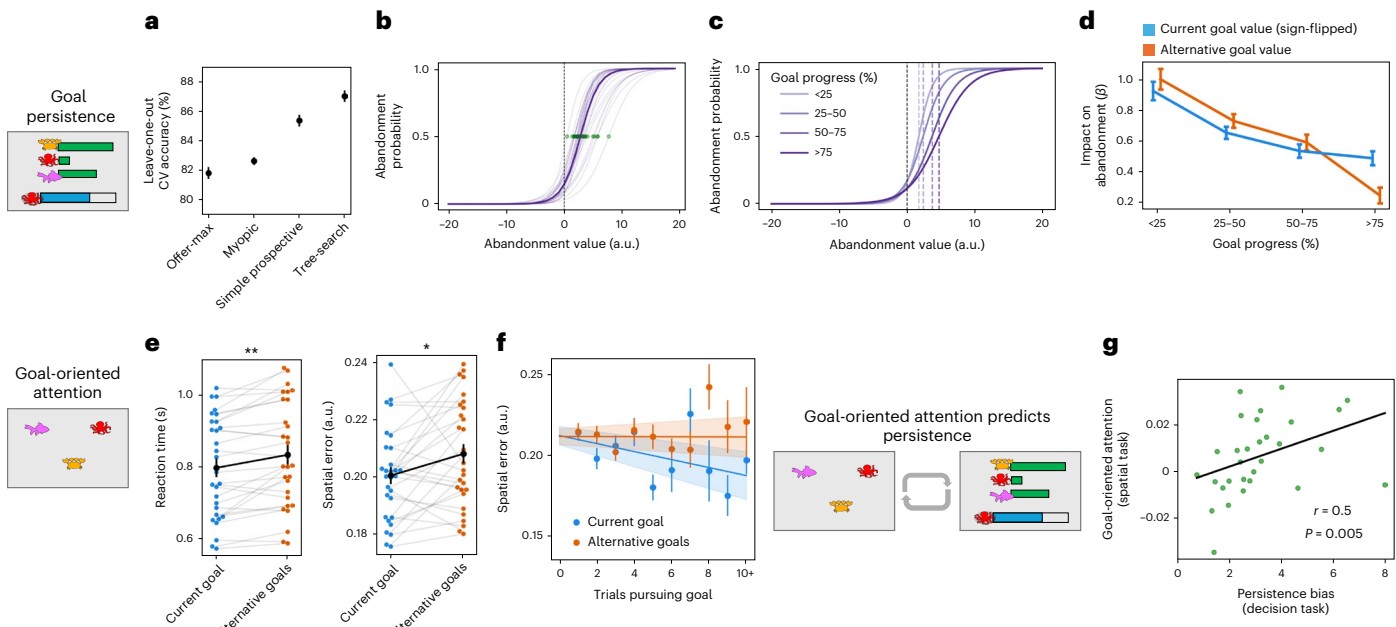

**Fig. 2 | Behavioural results. a–d**, Decision task. **a**, The tree-search model (approximating optimal behaviour) captured choices better than simple heuristic models. Mean ± s.e.m. of cross-validation accuracy ($n = 30$ participants). CV, cross-validation. **b**, Probability of goal abandonment as a function of the tree-search value of abandonment. Although the tree-search model captured choices best, people showed an additional bias towards persisting. Bold line shows fits across all participants; transparent lines show individual participants ($n = 30$). Green dots indicate indifference to abandonment, used as an index of individual persistence biases. **c**, Across individuals, persistence biases increased with goal progress (that is, proportion of the net completed). Successive purple lines show probability of abandonment as a function of tree-search abandonment, binned by goal quartile (shown for illustration). **d**, Over the course of goal pursuit, the impact of temptation (attractive alternatives) disappeared more than the impact of frustration (collapse in the current goal value). Blue and orange lines depict the influence of current goal value and (sign-flipped) best alternative goal value on abandonment choices across goal pursuit. Mean ± s.e.m. of beta weights ($n = 30$ participants). **e,f**, Attention task. **e**, In the interleaved spatial task, both reaction times (left) and memory error (right) were lower for the current goal stimulus (blue) compared with alternative goal stimuli (orange). Mean ± s.e.m. of RT and error are plotted ($n = 30$ participants); stars indicate two-sided paired $t$-test; RT difference: $t_{(29)} = 3.30, P = 0.003$; error difference: $t_{(29)} = 2.25, P = 0.032$. **f**, As participants invested more trials in a particular goal, spatial error decreased for the goal stimulus (blue) but not for alternative goal stimuli (orange). Mean error is plotted against trials pursuing the goal; dots show binned means ± s.e.m., with added regression lines (shaded region indicates s.e.m. of regression lines across participants; $n = 30$ participants). **g**, Relationship between decision and attention tasks. Individuals showing greater goal-oriented attention had higher persistence biases (Spearman's $r = 0.50, P = 0.005$, two-sided, 95% CI = (0.17, 0.73)). Spatial error enhancement for the current goal compared to alternative goals (from **e**, right) is plotted against persistence bias (from **b**, green). Persistence biases and attention biases come from separate testing session data (inside and outside the scanner, respectively).

(Fig. 2g; note that this relationship holds even when attention biases and decision biases originate from separate behavioural testing sessions; using persistence biases fit to data from scanner-only session: Spearman's $r = 0.50, P = 0.005$, two-sided, 95% CI = (0.17, 0.73); using persistence biases from data aggregated across both scanner and post-scan sessions: Spearman's $r = 0.53, P = 0.003$, two-sided, 95% CI = (0.20, 0.75)). This demonstrates that an individual's tendency to overpersist with the current goal is related to their allocation of selective attention towards the current goal.

**Goal abandonment due to 'temptation' versus 'frustration'**

How does progress towards a goal affect peoples' sensitivity to the value of switching away to an alternative? Pressure to abandon the current goal comes from two directions: an alternative good might become more attractive, pulling the agent towards the better option ('temptation') or the value of the goal good might collapse, pushing the agent away from the current goal ('frustration'; see Fig. 1b for example). Given that participants displayed increasing goal-oriented attention, we predicted that as a consequence, value associated with alternative goals would impact behaviour less than value associated with the current goal over the course of goal progress.

We found that people indeed showed an asymmetry in their use of these value sources compared with the optimal model, which developed during goal pursuit. To test this, we predicted abandonment choices in a regression model using the interaction between goal

progress and each source of value according to the tree-search model. Both sources of value impact behaviour less over the course of goal progress (alternative value × goal progress: two-sided $t$-test of beta weights against zero, $t_{(29)} = 7.97, P = 8.57 × 10^{-9}$, Cohen's $d = 1.48$; current goal value × goal progress: $t_{(29)} = 7.09, P = 8.48 × 10^{-8}$, Cohen's $d = 1.32$). However, this loss of influence on behaviour affected alternative goal value more than current goal value (difference between slopes: two-sided paired $t$-test, $t_{(29)} = 3.39, P = 0.002$, Cohen's $d = 0.63$, visualized in Fig. 2d by binning the data). In other words, over the course of goal pursuit, the impact of temptation from alternatives fades more rapidly than the impact of frustration with the current goal.

## fMRI results

### vmPFC activity tracks goal progress between decisions

Our behavioural analyses showed pervasive effects of goal pursuit on attention even outside the decision-making period. We reasoned that the brain regions involved in these attentional biases should similarly show goal-progress-related neural activity persisting outside the decision period. We conducted a whole-brain general linear model (GLM) analysis focusing on the intertrial period, modelling blood-oxygen-level-dependent (BOLD) activity using regressors capturing an individual's position in the goal (goal progress: proportion of target completed), the value of the current goal and the value of the best alternative in the previous trial (according to the tree-search model), and the decision itself (binary abandonment versus

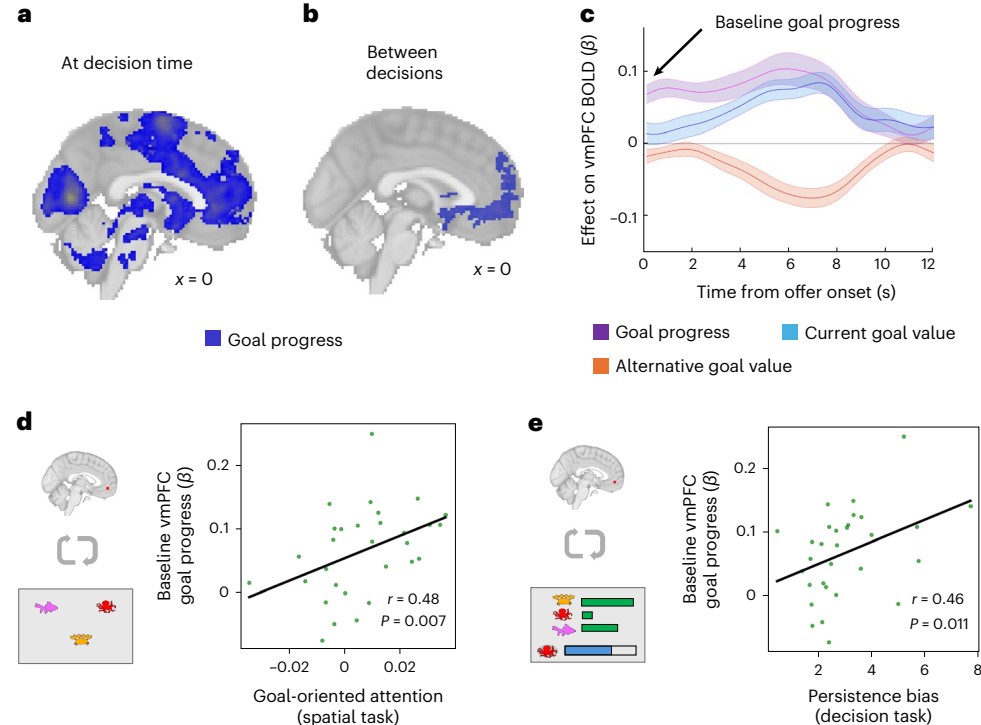

**Fig. 3 | Goal-related baseline vmPFC activity correlates with individual differences in behaviour. a,** Cluster-corrected activity representing goal progress time-locked to the onset of the decision period. **b,** Cluster-corrected activity representing goal progress time-locked to the intertrial fixation cross (see 'Whole-brain intertrial analysis' in Methods). While there was widespread activity in the occipital and parietal areas at decision time (**a**), the majority of these areas did not track goal progress 'between' decisions, where the highest peak was in the vmPFC. **c,** Time course of vmPFC activity at the onset of option offers, depicting the impact of goal progress (purple), current goal value (blue) and best alternative value (orange) at decision time (beta weight on BOLD activity). Error bars show s.e.m. across participants. We follow previous studies by defining baseline activity as the unconvolved neural activity at offer onset[23], which due to the haemodynamic delay captures 'pre-decision' activity rather than decision-related activity (Extended Data Figs. 7 and 8). **d,** Relationship

between baseline goal tracking in the vmPFC and goal-oriented attention (Spearman's $r = 0.48$, $P = 0.007$, 95% CI = (0.15, 0.72)). The baseline measure corresponds to the impact of goal progress on activity in the vmPFC ROI at the moment of choice onset (**c**). Goal-oriented attention refers to the accuracy advantage for remembering the current goal location compared to alternative goal locations in the spatial attention task (Fig. 2e, right). Note that the attention measure comes from a separate testing session outside the fMRI scanner. **e,** Relationship between baseline goal tracking in the vmPFC and persistence biases in the decision task (Spearman's $r = 0.46$, $P = 0.011$, 95% CI = (0.12, 0.70)). Notably, the relationships between neural activity and behavioural goal biases are specific to baseline activity in the vmPFC; baseline activity in other regions of interest and vmPFC activity in response to the decision itself are not predictive of behavioural biases (see Extended Data Fig. 8 for control analyses at key timepoints and regions).

persist choice; see Extended Data Fig. 6 and Methods for full details of neural GLM). In addition, we controlled for decision-related activity by adding all of these regressors at decision time (time-locked to the onset of offers; see Extended Data Fig. 7 for additional goal progress controls). We found that the peak of activity tracking goal progress during the intertrial period was in the vmPFC (Fig. 3b).

### Pre-decision vmPFC predicts differences in goal commitment

Previous studies have found that baseline vmPFC activity (activity before a decision) predicts biases or priors which affect subsequent decision-making[22–24]. As vmPFC tracks goal progress between decisions, we hypothesized that the strength of the baseline vmPFC signal (Fig. 3c) would predict the degree of commitment bias (unwillingness to switch goods) across individuals.

We extracted baseline activity on a trial-by-trial basis in our vmPFC region of interest (ROI) and quantified the extent to which pre-decision activity was tracking goal progress for each individual. We found that this baseline goal-related activity correlated with an individual's overall persistence bias during the decision-making task (Spearman's $r = 0.46$, $P = 0.011$, two-sided, 95% CI = (0.12, 0.70); Fig. 3d; see control analyses in Extended Data Fig. 8).

If baseline vmPFC activity also reflects the degree to which attention is oriented towards the current goal, we reasoned that it should also correlate with differences in goal-directed attention in the second,

decision-free task. This was indeed the case: across participants, the strength of the baseline goal–progress signal in the vmPFC predicted greater accuracy for the current goal relative to alternative goals in the attention task (Spearman's $r = 0.48$, $P = 0.007$, two-sided, 95% CI = (0.15, 0.72); Fig. 3e). This was particularly striking as the spatial decision-free task was carried out in a separate session outside the scanner.

### Neural activity related to goal pursuit at decision time

We investigated neural activity at the time of the decision in a whole-brain analysis (regressors time-locked to the onset of the offers). The decision-time analyses revealed a much broader network of areas sensitive to goal pursuit. Specifically, as an individual progressed towards completing the goal, activity in a wide range of areas increased, including the medial prefrontal cortex, striatum and cingulate areas, as well as large regions of the occipital and parietal cortices ('goal progress' regressor; Fig. 3a). Note that while this wide range of neural areas tracked goal progress during the decision, only a subset of areas focused on the vmPFC continued to track progress during the intertrial period as previously described (Fig. 3a,b).

In addition, we found value-related activity consistent with previous studies engaging brain networks in choices between staying with a default versus switching to an alternative. Both medial prefrontal cortex and striatum increased their activity as the value of

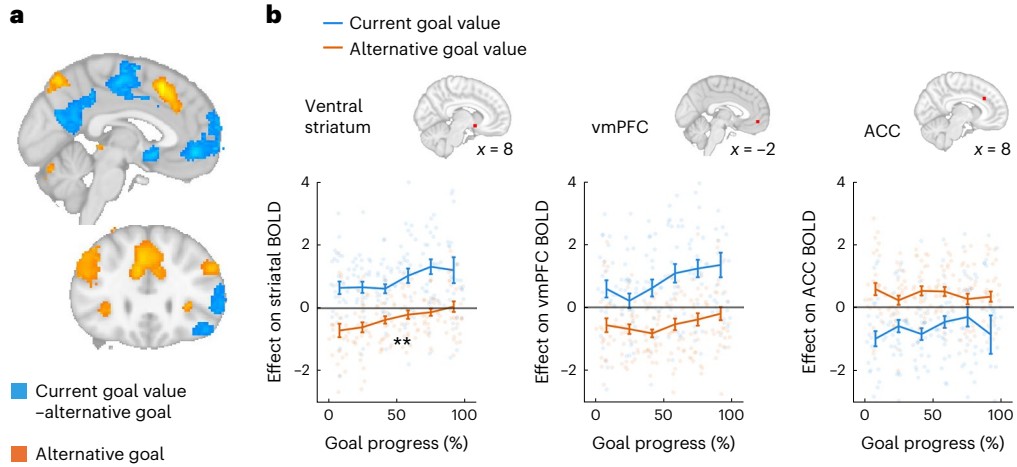

**Fig. 4 | Neural activity related to the value of persistence and abandonment. a**, Results from the whole-brain analysis showing cluster-corrected peaks for the contrasts of current goal value–best alternative value (blue) and best alternative value–current goal value (orange). **b**, Modulation of value-related activity in ROIs over the course of goal pursuit, where the red dots in the brain images indicate the ROI location for activity shown in each plot. Here we show the effect of value on the BOLD signal (beta weight) as a function of the proportion of the goal completed, binned for illustration. Blue shows the impact of current goal value, while orange shows the impact of alternative goal value. Mean beta weight ± s.e.m. are depicted, while dots show beta weights for individual participants ($n = 30$ participants). In the striatum (left), there was a significant reduction in the representation of alternative goal value across goal progress (orange line; stars indicate significant interaction between alternative goal value and goal progress; Wilcoxon signed rank, $Z = 2.37$, $P = 0.009$, $n = 30$), parallel to the reduction in sensitivity to alternative goals seen in behaviour. In contrast, representations of the current goal value were maintained throughout goal pursuit in all ROIs.

persisting with the goal increased (value of current goal–value of best alternative; Fig. 4a, blue; time-course of vmPFC value-related activity shown in Fig. 3c). In contrast, the ACC, presupplementary motor area (preSMA), bilateral dorsolateral prefrontal cortex (dlPFC) and bilateral insular all showed the opposite profile: activity increased as the value of abandonment increased (value of best alternative–value of current goal; Fig. 4a, orange) and activity was higher on trials where the participant chose to abandon the current goal (Extended Data Fig. 6b; see Extended Data Table 1 for activity peaks). We included response times as an additional control regressor, previously argued to be a proxy for choice confidence[33,34]. We found that ACC activity was also higher when participants were slower to respond, but we found no relationship between response times and vmPFC activity (Extended Data Fig. 6e).

### The striatum shows decreasing sensitivity to alternative goals
As attention to the current and alternative goals varies with goal pursuit, we should expect to see changes in neural representations of these goals. In particular, in behaviour we observed an intriguing asymmetry, namely, that as goal commitment increased, sensitivity to alternative goal value ('temptation') was reduced more than sensitivity to the current goal value ('frustration'). We therefore asked how value signals relating to the current and alternative goals change as a function of goal pursuit.

Parallel with our behavioural results, we found an asymmetry between how goal pursuit affected signals relating to alternative and current goal value in the ventral striatum. Specifically, representations of alternative value disappeared in the ventral striatum over the course of goal pursuit, but activity continued to covary with the current goal value (Fig. 4b, left; interaction between best alternative value and goal progress: Wilcoxon signed rank, $Z = 2.37$, $P = 0.009$, $n = 30$, one-sided; interaction between current goal value and goal progress: Wilcoxon signed rank, $Z = -1.03$, $P = 0.152$). This mirrored the behavioural finding that people became relatively less sensitive to temptation by alternative goods, while maintaining sensitivity to the value of the chosen goal, over the course of goal pursuit. In contrast, there was no significant change in the representation of alternative value over goal pursuit in either vmPFC ($Z = 1.19$, $P = 0.116$) or ACC ($Z = 0.39$, $P = 0.348$).

## Lesion patient study
### Damage to vmPFC reduces commitment bias for the current goal
Taken together, the behavioural and fMRI results suggest that the vmPFC maintains attention to the chosen goal, leading to over-persistence or an unwillingness to switch goals. To test the causal nature of this association, we conducted an independent study using the same paradigm, with a sample of 23 participants with brain lesions in variable locations (see Fig. 5a for map of lesion overlap across patients). We focused on persistence bias, defined as the tendency to persist with the chosen goal beyond the point at which it would be optimal to switch, as the key behavioural marker of goal commitment.

We began by investigating whether damage to particular areas reduced persistence in the lesion patient group, independent from any priors from our fMRI study. We asked at what locations damage predicted a reduction in persistence bias by running a voxelwise regression analysis using damage in each voxel (binary regressor) to predict persistence bias. Independently corroborating the findings of our fMRI study, the only region where damage predicted a reduction in persistence bias was an area in the vmPFC (Fig. 5b, green cluster).

We then asked how much the region identified in our lesion patient study aligned with the findings of our fMRI study. Our fMRI study had identified a subset of areas carrying signals relating to goal pursuit even between decisions, focused on the vmPFC. We split all patients into two groups on the basis of whether they were damaged within a region of interest at the peak of this fMRI activity, found in the vmPFC (region of interest centred on the peak of the activity tracking goal progress during the intertrial interval (ITI) in our fMRI study; shown in Extended Data Fig. 9d). There were four lesion patients with damage to this region of interest, and this group had reduced persistence biases compared with patients with damage elsewhere and with age-matched healthy controls (Fig. 5c). We found that these four patients who had damage within the region pre-defined by our fMRI study corresponded to four (out of the five total) patients identified from our independent voxelwise patient analysis. Therefore, our fMRI study and lesion patient study independently converge to identify the same vmPFC region as being relevant for goal commitment.

Next, we ruled out the possibility that the vmPFC-damaged group were simply performing worse in some general way, for example, by

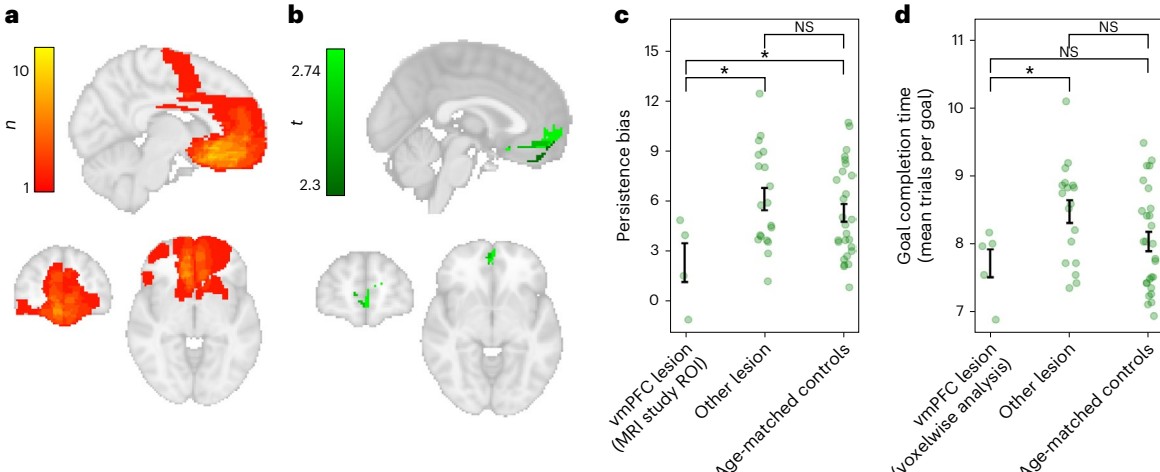

**Fig. 5 | Lower goal commitment in patients with vmPFC lesions. a,** Lesion overlap maps of the 23 patients who took part in the study (maximum overlap in a voxel was 10 participants). **b,** Results from the whole-brain voxelwise analysis. Green shows areas where lesion damage predicts lower persistence biases. Above-threshold *t*-statistics (*t* > 2.3 before cluster correction) are displayed for illustrative purposes. We controlled for multiple comparisons by performing cluster correction using FDR described in Methods. Only the vmPFC cluster survived whole-brain cluster correction (cluster threshold *t* > 2.3 (*P* < 0.01, one-sided), cluster size = 269 voxels, threshold cluster correction size = 255 voxels, cluster peak = (0,42,−14), *t*-statistic at cluster peak = 2.74, *n* = 5 patients with damage within cluster). **c,** Patients with damage to the vmPFC region identified in the fMRI study show reduced persistence bias. Patients were split into two groups depending on whether they were damaged within a region of interest centred on the peak of BOLD activity tracking goal progress between decisions in healthy participants. This area was damaged in 4 patients, corresponding to 4 out of the 5 patients independently identified in the voxelwise analysis in **b**. Patients

with damage to this region showed lower goal commitment than patients with lesions elsewhere and age-matched controls (one-sided permutation test for lower persistence biases in vmPFC-damaged group compared with other lesion patients: difference in means = 3.79, *P* = 0.012; lower persistence bias in vmPFC-damaged group compared with age-matched controls: difference in means = 2.97, *P* = 0.023). Error bars show s.e.m. in each group; green dots depict individual biases. **d,** Post hoc analysis showing that patients with damage within the identified vmPFC region from the voxelwise analysis shown in **b** performed better than other lesion patients and no worse than age-matched healthy controls. Performance was measured as the average number of trials to complete a goal, where lower scores correspond to faster goal completion (one-sided permutation test for faster performance in the vmPFC group compared with other lesion patients: difference in means = 0.76, *P* = 0.015; faster performance in the vmPFC group compared with age-matched controls: difference in means = 0.32, *P* = 0.190, not significant (NS)).

making random choices or forgetting the goal. An important point to note is that, because participants in general overpersist, a reduction in persistence biases should actually lead to an improvement in task performance, if participants switch goals at points at which it is beneficial to do so (rather than making random switches due to, for example, task disengagement). This is exactly what we found: the five vmPFC-damaged patients identified in our voxelwise analysis in fact performed significantly better than patients with damage elsewhere and no worse than age-matched healthy controls (Fig. 5d; performance was quantified as mean trials to fill a net, so smaller values indicate goals are completed faster).

Finally, we used further post hoc analyses to verify that (1) vmPFC patients were not responding more stochastically and (2) vmPFC patients were not using a different normative model to solve the task. We formally quantified stochasticity as inverse temperature and found that the vmPFC group showed no difference in inverse temperature compared with other patients or age-matched controls (Extended Data Fig. 9b; see recoverability of inverse temperature parameter in Extended Data Fig. 4c). We also found that, similar to the MRI participants, decisions for all three groups in our lesion study are best described by the tree-search model (Extended Data Fig. 9a), suggesting that vmPFC patients were not using a simpler response heuristic.

Taken together, these results suggest that patients with damage to this region of the vmPFC are not simply using a different task strategy or responding more randomly, but instead are less biased towards overpersisting with a goal.

## Discussion

Many rewards are only obtained after a period of persistent effort. Therefore, a key challenge for agents is to maintain a balance between commitment with the current goal and flexibility if it ceases to be worthwhile. The current study presents evidence that the shift towards

goal commitment is supported by the vmPFC and relates to mechanisms of goal-oriented selective attention. It is well known that people tend to overpersist with chosen goals (the 'sunk-cost' fallacy). Rather than representing persistence biases as a (perhaps irrational) factor in the decision process itself, we argue that it is better understood in terms of a more pervasive attentional effect: mechanisms of selective attention, mediated by the vmPFC, prioritize information related to the current goal over alternative goals, resulting in reduced sensitivity to attractive alternatives ('temptation'). This attentional bias is sustained in time and generalizes outside the decision context, as participants showed reduced sensitivity to sensory features of goal-irrelevant stimuli (such as their location in space), particularly as the goal state is approached.

We developed a pair of complementary tasks to measure how attentional and decision-making biases develop together during goal pursuit. In the decision-making task, commitment to a goal is required to realize rewards, but to perform well at the task, participants must also remain sensitive to changes in the value of the current and alternative goals. Participants tended to persist with goals longer than was optimal. As people progressed towards the goal, they became less sensitive to the value of alternative goals compared with the value of the current goal, suggesting an increasing focus of attention on the current goal as they neared goal completion. We further probed this attentional account by interleaving the decision task with an unrelated and decision-free spatial working-memory task. We found that participants were better able to recall the location of stimuli associated with the current goal and this tendency increased as they continued longer with the goal. Furthermore, there were stable individual differences in persistence with a goal, which were predicted by individuals' sustained goal-directed attention outside the decision period. Individuals who were more biased to persist with a goal showed higher goal-oriented

selective attention, even when these metrics were captured in separate testing sessions and an unrelated, decision-free task.

We present multiple converging lines of evidence demonstrating that the vmPFC plays a key role in this process. First, our fMRI study found that the vmPFC carries sustained goal-related information between decisions in our task, and baseline activity before the decision predicts the two independent behavioural metrics of goal capture: both an individual's bias to persist with the current goal and their bias to prioritize goal-related stimuli in attention. This was the case despite the fact that attention was measured during a separate task outside of the scanner. Second, we show that the vmPFC is causally involved in goal commitment: patients with damage to the same region have reduced biases to persist with the current goal.

In various contexts, the medial prefrontal cortex has been shown to support the selection of goal-relevant information[35–37], flexibly adapting to changes in the current goal[15–19] possibly through compression of goal-irrelevant information[21]. Other studies have also linked vmPFC activity to visual attention specifically, both in responding to exogenous manipulations of attention[38,39] and in mediating visual attention[40]. Here we present results bringing together these distinct bodies of research, suggesting that the role the vmPFC plays in selecting goal-relevant information is linked to visual attention.

We find that across healthy individuals, baseline vmPFC activity (activity before a decision is made) predicts both decision and attention biases in our task. This builds on growing evidence in both monkeys and humans finding that baseline vmPFC activity plays a role in influencing how options are processed and subsequently, which choice is made[22–24]. Baseline vmPFC activity has been argued to bias upcoming choices in line with prior contextual factors, including both stable preferences (such as tastes in music or food types[22]) and dynamic states (such as satiety or mood[23,24]). Our results provide evidence that another dynamic state, namely, pursuit of a chosen goal, modulates behaviour through baseline vmPFC activity. We argue that our results also offer a possible mechanism for these effects: sustained vmPFC activity could drive global changes in top-down attention, affecting how options are processed and which decision is subsequently made.

In theory, the vmPFC could vary with goal-relevant information without playing any causal role in the decision process. To test the causality of vmPFC activity in goal persistence, we carried out an independent study using the same paradigm with 23 lesion patients. Through a voxelwise analysis of damage in our patient sample, we identified a vmPFC cluster in which damage predicted reduced persistence biases. The area identified in patients closely corresponded to the area involved in persistence among healthy individuals, providing striking evidence that vmPFC plays a causal role in goal commitment. Our results expand on previous reports that lesions to this area in both humans and primates interfere with the ability to prioritize relevant decision variables, for example, in cases when a distracting alternative is introduced[41,42], or an option has been de-valued[43].

While previous lesion studies have found this patient population to behave more stochastically[41,44], notably lower persistence biases among vmPFC lesion patients in our task cannot be explained by an increase in stochasticity. In fact, we find that patients with vmPFC damage performed better than other lesion patients and no worse than age-matched controls. In a goal-pursuit context, healthy individuals may have a tendency to overconstrain the decision space by focusing only on the current goal and ignoring alternatives. In contrast, a lesion to this area of vmPFC may reduce selective attention to the goal, allowing alternatives to maintain their relevance throughout goal pursuit. We note that, while this is beneficial in our task, it is likely to be advantageous to constrain the task space in ecological goal-pursuit settings, both in terms of optimal neural resource allocation (that is, attending to goal implementation and avoiding cognitive switch costs) and in structuring behaviour over time.

Our results also reveal how neural value representations change dynamically across goal pursuit, consistent with attentional prioritization of the current goal. We found that late in goal progress and compared with an optimal model, people demonstrated reduced sensitivity to the value of alternative goals compared with the value of the current goal. When the value of alternatives lost influence over behaviour, this was mirrored by a reduction in the representation of alternative value in the ventral striatum over goal pursuit. While we are not aware of other studies showing this pattern, the ventral striatum is known to respond to goal pursuit, for example, through striatal dopamine ramps during goal approach[45,46].

We found that both the ACC and dlPFC positively covaried with the value of abandonment, and were more active when participants chose to abandon. This is consistent with previous work showing that activation in these areas and in the ACC in particular represents the value of alternative options[27] and is more active when an individual disengages from the present action[32,47] or explores the environment[19,31]. In fact, when people switch out of an exploitative state towards exploration, ACC activity predicts changes in task representation in the vmPFC[48]. While the vmPFC represents the current goal and enables goal commitment, the ACC is likely to underpin behavioural flexibility during goal pursuit by consistently tracking other options. In contrast to the striatal effects, we found relatively sustained representations of alternative options throughout the goal in the ACC, supporting previous studies showing that the ACC drives flexibility. The fact that people show increasing biases to persist rather than remain flexible could be explained by increasing dominance of regions such as the vmPFC over the ACC. We note that the vmPFC-lesioned patients in this study made effective abandonment choices that allowed them to perform well in the task. While the vmPFC contributes to persistence biases, it does not seem necessary for making appropriate abandonment choices, which are likely to depend on other areas such as the ACC.

There are limitations to how the attentional effects are interpreted in this study. The goal-directed attentional biases observed could stem from either memory-guided 'top-down' orientation towards goal-relevant information[49,50] or 'bottom-up' attentional capture from the high-incentive salience of goal-related stimuli[51–53]. While our task cannot definitively distinguish between these possibilities, one relevant consideration is how quickly attention shifts towards the new goal stimulus, rather than reflecting the reward history of stimuli across the study. This rapid attentional adaptation to new goals may be evidence in favour of goal-directed attentional orientation (although some studies have found goal congruency to be stronger predictors of attentional capture than long-run value[37,54]). The top-down attention interpretation is also consistent with previous evidence of vmPFC involvement in memory-guided attention[50,55].

We have argued that activity in the vmPFC relates to persistence with a goal. However, previous studies have shown that vmPFC activity is also related to decision confidence[33,56]. Since choices to persist with partially completed goals tend to be associated with higher confidence than choices to start again with an alternative goal, it is important to consider whether this could be a potential confound. Using response times as a proxy for confidence, we did not find any relationship between vmPFC activity and response times, suggesting that vmPFC activity is not obviously related to simple measures of decision confidence in our task. One possible explanation for this disparity with previous findings could relate to the specific setting of incremental goal pursuit. If there is a strong default to persist with the current goal during goal pursuit, vmPFC activity may not track trialwise variations in confidence associated with offers unless a drastic change prompts re-evaluation of the goal.

Our study suggests that goal pursuit leads to global changes in how the environment is processed, implemented through alterations in vmPFC activity. Goal-directed selective attention provides a mechanism by which animals can prioritize goal completion while remaining sensitive to exceptional alternatives, since attentional selection itself

can be graded[14]. While goal commitment may manifest in seemingly irrational tendencies to persist with a previous decision, the ability to filter information to prioritize a selected task would be critical in ecological settings.

## Methods

### MRI study

**Participants.** A total of 31 participants (19 female; mean age 25 years, normal or corrected-to-normal vision) were recruited via email circulation on Oxford University mailing lists and social media. One participant was excluded from the recruited sample because they opted out of the study before the MRI scan, leaving a total of 30 participants whose data are analysed in this study. No statistical methods were used to pre-determine sample sizes, but our sample sizes are similar to those reported in previous publications[18,19,30]. Ethical approval for the fMRI study was obtained from the Oxford Central University Research Ethics Committee (Ref: R72921/RE001). This study was not pre-registered. All participants gave written informed consent before the experiment. Participants were paid £15 per hour plus a performance-dependent bonus of £8–12.

**Experimental procedure.** The training, scan and post-scan task were all carried out in a single session lasting 2.5–3 h in total. Before the fMRI scan, participants were trained on the task for ~20 min. Participants practiced on three full example blocks (on average ~25 trials, dependent on performance) with the interleaved spatial attention task included, and one additional example block without the spatial attention task (scanner version). Comprehension questions were included at the end of training to ensure that participants had understood the task structure. Once this had been verified, participants entered the scanner and completed 300 trials of the decision task only (since the spatial task could not be performed with the button box inside the scanner), lasting 50–60 min (scanner session). Participants then completed the spatial variant of the task for an additional 100 trials outside of the scanner, lasting 20 min (post-scan session). Once the post-scan session was complete, participants filled out a short debrief questionnaire. The experimental task paradigm was created using PsychoPy (v.2021.1.2).

**Primary decision task.** Participants were told their aim was to fill as many nets with seafood as possible across the study, limited only by the number of choice trials in the study. The number of trials remaining in which the participants could continue to fill nets was shown in the top right corner of the screen throughout the study (Extended Data Fig. 1a). Above the indication of trials remaining, the number of points earned (nets completed so far) was shown, where each completed net was converted to a £0.25 bonus payment at the end of the study.

At the start of each block, participants were shown the size of the net to be filled as an empty grey bar at the bottom of the screen (Extended Data Fig. 1b). Blocks ended when a net was complete and a point was won (Extended Data Fig. 1c). On each trial, participants were presented with three offers associated with the three sea creatures (always crab, octopus and fish). Offers were shown as horizontal coloured bars on the screen next to their respective creature, where the size of the bar translated exactly to the quantity which would be added to the net if that creature was chosen. Offers were mostly positive (indicated by green bars) but could sometimes become negative (indicated by a red bar). If a negative offer was selected, the quantity of the bar would be subtracted from the net. Once a net was empty, nothing more could be lost so choosing a negative offer would lead to no change.

In the scanner, participants indicated which creature they wanted to accumulate using a button box where the first three buttons corresponded to the top, middle and bottom creatures on the screen. Outside the scanner, participants selected the creature by clicking with the mouse. Note that across all versions of the task, the horizontal order of the three creatures on the screen was randomized on every trial to avoid confounding persistence with motor perseverance. Once the creature was selected, the participant viewed the net being updated according to their choice.

**Spatial variant.** After completing the task for 300 trials inside the scanner, participants performed 100 trials of a spatial variant of the task outside the scanner. The spatial variant included an interleaved spatial attention task before every decision (Extended Data Fig. 1e). Participants viewed the three creatures flashed up simultaneously for 500 ms in randomized locations across the screen. Participants were then probed in a random order on the location of each creature. When the icon of each creature appeared in the top right corner of the screen, participants responded by using their mouse to click on the location at which they remembered it appearing. While it was not possible for participants to perform the spatial attention task inside the scanner (due to the impracticality of reporting three spatial locations on every trial with a button box), we matched the basic structure of the scanner variant to the spatial variant by having participants passively view the three creatures flashed on the screen during an ITI of between 2.5 and 8 s (Extended Data Fig. 1d).

**Schedules.** *Schedule generation procedure*. For each block, the size of the net and the option offers differed. The net sizes were drawn from a uniform distribution (minimum 12, maximum 72). The initial values for the three options were drawn independently from a normal distribution at the start of each block (mean = 6, $\sigma^2 = 1$). From trial to trial, the offers for each option changed according to independent Gaussian random walks ($\sigma^2 = 0.8$). In addition, on each trial there was an independent probability of any option changing more drastically in its associated offer ($P = 0.1$ jump up, $P = 0.1$ jump down), corresponding to an option becoming substantially more 'bountiful' or 'scarce' for fishing opportunities. The jump function consisted of drawing a random value between 3 and 9 points higher or lower than the option's starting offer, which corresponded to the new offer for that item. After a jump, the subsequent offers for that option would continue to change according to a random Gaussian walk from the new starting location (see Fig. 1b for example trajectories created using this procedure). To select pairs of net sizes and option offers for which completing the net was non-trivial yet feasible, we chose combinations where goals were completed in more than 3 trials and less than 15 trials when choice behaviour was simulated using the tree-search model.

*Schedule variants*. To minimize schedule-specific artefacts, we generated 5 different schedules which each consisted of 45 blocks of 100 trials. A block ended when the net was filled, so participants on average viewed only 7 trials per block before completing the net. For each MRI participant, separate schedules were randomly selected for the within-scanner and post-scanner sessions. In the lesion patient study, the same schedule was used across all individuals (including age-matched controls) due to the limited sample size for lesion patients. Each session ended after a predetermined number of trials (300 in the MRI scanner, 100 in the post-scan session and 250 for all participants in the patient study), so no participant was able to complete all 45 blocks of a schedule within the available experimental trials.

**Behavioural models.** We investigated participants' choice strategy by fitting their behaviour to a set of possible models capturing different heuristics or strategies. Four models with increasing complexity were tested as candidates for describing peoples' subjective evaluation of the offers (see Extended Data Fig. 3a for a graphic depiction of the strategies):

1.  Offer-max model: The agent chooses the largest offer on screen, regardless of the accumulated contents in the net. The values of the three items according to the model are equivalent to the current offers for each item.

2. Myopic model: The agent maximizes accumulated value on the current trial. This means they will only switch if an alternative offer is greater than the combined contents of the net and the offer for the current goal item. The value of the goal item is equal to the accumulated value plus the goal item offer, while the value of the alternatives is simply equivalent to their current offers.

3. Simple prospective model: The agent calculates how much progress towards the goal each offer will entail, where progress is the proportion of the remaining unfilled net that will be completed after choice. Mathematically, the value of an option according to this model is the current offer for each option, divided by the quantity of net left to fill (when choosing that option). Intuitively, this model values each option on the basis of the number of trials needed to fill the whole net, if the option values stay constant throughout.

$$\text{Goal good value} = \left\lceil \frac{\text{Option offer}}{\text{Net size} - \text{Accumulated value}} \right\rceil \quad (1)$$

$$\text{Alternative good value} = \left\lceil \frac{\text{Option offer}}{\text{Net size}} \right\rceil \quad (2)$$

A central difficulty for a model that estimates value in this way is dealing with negative offers. Negative offers would reverse the respective values, meaning that implausibly, negative offers associated with the goal good are valued less than negative offers associated with alternative goods. To address this problem, we set the value of negative offers associated with alternatives to their raw (negative) offer, and the value of negative offers associated with the goal option to the proportion of progress they would be losing, that is, the offer divided by the accumulated value.

4. Stochastic tree-search model: The agent uses information about offer trajectories to simulate possible futures for the different candidate options, choosing the option that is forecasted to complete the net fastest. Specifically, it samples possible future trajectories for the three options and calculates each option's value as the (negative) average number of trials until net completion across the iterations (if it were chosen on this trial).

The same statistics used for creating the experimental offers were used when the model simulated the future trajectories of the options (procedure described in 'Schedule generation procedure'). In other words, this model possesses task knowledge of how offers are likely to change over time and leverages that to compute a better estimate of how long each option will take to fill the net.

We verified that behaviour from the different models can be distinguished in the task schedules used (full details of the model validation process are included in Supplementary Information).

**Model fitting.** Participant choices were aggregated across the scanner session (300 trials) and post-scanner session (100 trials) before model fitting. In each case, the model value of switching was calculated as the model's value for the current goal subtracted from the model's value for the best alternative goal. To determine the best-fitting normative model, we fit the following models to behaviour:

1. $\text{SV}_{\text{abandon}} = \beta_0 + \beta_1(\text{alternative value}_{\text{offer-max}} - \text{goal value}_{\text{offer-max}})$
2. $\text{SV}_{\text{abandon}} = \beta_0 + \beta_1(\text{alternative value}_{\text{myopic}} - \text{goal value}_{\text{myopic}})$
3. $\text{SV}_{\text{abandon}} = \beta_0 + \beta_1(\text{alternative value}_{\text{prospective}} - \text{goal value}_{\text{prospective}})$
4. $\text{SV}_{\text{abandon}} = \beta_0 + \beta_1(\text{alternative value}_{\text{tree-search}} - \text{goal value}_{\text{tree-search}})$

where $\beta_0$ is the intercept and $\beta_1$ is the slope capturing the use of model value. SV refers to subjective value. We fit these models in a mixed effects logistic regression analysis predicting abandonment choices, where the intercept and slope were also modelled as random effects across participants.

We used a leave-one-out cross-validation process to evaluate between models since the models differed in their conceptual complexity but not in the number of fitted parameters. For each participant, we fit each of the mixed-effects models to the choices of all other participants ($n = 29$). For the held-out participant, we then computed the predicted abandonment value for each trial and transformed this into the predicted probability of switching using the softmax function:

$$P_{\text{abandon}} = \frac{1}{1 + e^{-\text{SV}_{\text{abandon}}}} \quad (3)$$

We took the absolute difference between the predicted probability of switching and each held-out participant's true responses, and subtracted this difference from 1 to compute the model accuracy for each participant separately. This allowed us to evaluate both the overall cross-validation accuracy of each model and the frequency of best-fitting models across participants (Extended Data Fig. 3e–g).

Data were analysed in Python (3.11.5) and MATLAB (v.R2021_a). The following Python packages were used for data processing, analysis and visualization: pandas (2.1.1), NumPy (1.26.0), seaborn (0.12.2), Matplotlib (3.8.0), SciPy (1.11.2), statsmodels (0.14.0), Pingouin (0.5.3), rpy2 (3.5.11), Nilearn (0.10.2) and NiBabel (5.1.0).

**Persistence bias.** Once we established that the tree-search model was the best-fitting model of participant behaviour, we used this model to further investigate individual differences in persistence deviating from the model. We fit the logistic regression model predicting switches using tree-search value to each participant separately. A participant's indifference point (IP) is the model value of abandonment at which a participant is equally likely to persist or abandon (the 'shift' on the sigmoid function). Mathematically, this is equal to:

$$\text{IP} = \frac{-\beta_0}{\beta_1} \quad (4)$$

where $\beta_0$ and $\beta_1$ refer to the intercept and slope, respectively, from the logistic regression predicting participant abandonment choices from the model value of abandonment. Since persistence biases violated tests of normality, we used the one-sample Wilcoxon signed-rank test to determine whether the indifference points were significantly above zero.

Throughout subsequent analyses, the IP parameter fitted to each participant is referred to as their 'persistence bias' (that is, the bias towards persisting compared to the tree-search model). Persistence biases were recoverable in the empirical range and had good test-retest reliability across sessions. Extended Data Fig. 4a shows recoverability and test-retest reliability, and a full description of the parameter recovery procedure and test-retest analyses are included in Supplementary Information.

In addition, we investigated whether persistence biases were modulated by goal progress (defined as the proportion of the current net completed). Full details of goal progress analyses are included in Supplementary Information.

**Spatial task analyses.** The spatial task results come from a separate behavioural testing session after the fMRI session, where participants performed the same decision task with the addition of an interleaved spatial attention task before making each decision (the 'spatial variant' described above). We used this task to measure the relative distribution of attention between stimuli associated with the current goal and stimuli associated with alternative goals, across goal pursuit. We quantified spatial error as the Euclidian distance between the location of the participant's click and the true location at which the stimulus appeared, in normalized screen units. We quantified reaction times (RT) as the time (in seconds) between when a stimulus was probed

(appearing in the top left corner of the screen) and when the participant indicated their response.

We then categorized responses according to whether the probed stimulus was the current goal good or one of the alternatives. We excluded the first trial of every block from analyses, where no goods had yet been accumulated. Since the distribution of mean reaction times and mean error did not violate assumptions of normality, we used *t*-tests to determine whether mean error differed as a function of the status of the stimulus (that is, whether the stimulus was the current goal item or an alternative goal item).

We then investigated whether the spatial error bias developed as a function of goal pursuit (Fig. 2f). We fit two linear models for each participant, predicting (1) current-goal stimulus error and (2) alternative-goal stimuli error using the number of trials participants had been pursuing the goal, in each case modelling error using the following linear regression:

$$\text{Error} = \beta_0 + \beta_1 \text{Trials invested} \tag{5}$$

Since the beta weights did not violate assumptions of normality, we used two-sided *t*-tests to determine whether the $\beta_1$ coefficients across participants differed from zero (showing that error is dependent on the number of trials invested) for either the current-goal stimulus or the alternative-goal stimuli. We also tested for the difference between slopes, using a *t*-test to determine whether trials invested affected error differently for the current-goal versus alternative-goal stimuli.

Finally, we investigated whether goal biases in the spatial task were related to persistence biases in the decision task. To capture an individual's goal-oriented attentional bias, we took the difference between an individual's mean error for the current-goal stimulus and their error averaged across the two alternative stimuli (Fig. 2e). We tested for a relationship between an individual's goal-oriented attentional bias and their persistence. Spearman's correlation was used because as previously noted, persistence biases violated assumptions of normality.

**fMRI acquisition.** The fMRI data were collected at the Oxford Centre for Human Brain Activity using a 3T Siemens scanner with a multiband accelerated echoplanar imaging sequence with the following parameters: voxel resolution $2.4 \times 2.4 \times 2.4$ mm$^3$, repetition time = 1,230 ms, echo time = 30 ms, flip angle = 60°, field of view = 240 mm, multiband acceleration factor = 3, PAT factor = 2, encoding direction = PA. A tilt angle of 30° was used to minimize signal dropout in the orbitofrontal cortex[57]. Data were collected in two consecutive runs of ~25 min, where participants stayed in the scanner between runs.

**Pre-processing and analysis structure.** Data were preprocessed using FMRIB's Software Library (FSL), using the FEAT software tool[58]. Functional data were motion corrected using rigid body registration to the central volume[59,60]. Gaussian spatial smoothing was applied with a full-width half-maximum of 5 mm, and high-pass temporal filtering was applied with a cut-off of 60 s. Cardiac and respiratory data were processed using FSL's Physiological Noise Modelling (PNM) tool to model the effects of physiological noise in the MRI data[61]. Since participants completed the MRI session in two runs, parameter estimates were first estimated at the level of run (first level), then combined within individuals as fixed effects (second level), and finally combined across subjects using FMRIB's Local Analysis of Mixed Effects (FLAME1 + 2; third level)[62]. Multiple comparisons were corrected for using a *Z*-statistic threshold of 3.1 and a cluster probability threshold of *P* = 0.05.

**Univariate fMRI analyses.** *Decision-time analysis.* A GLM was used to model BOLD activity in pre-whitened data space. Seven regressors of interest were included in the main GLM, predicting BOLD activity at the onset of the decision period (all modelled as stick functions). These regressors included whether the choice on this trial was to persist or

abandon (coded as 1/−1), the tree-search value of the current goal, the tree-search value of the two alternatives, goal progress, goal size and response time. Since goal progress is correlated with tree-search value but our behavioural analyses show that it is an additional predictor of abandonment beyond tree-search value (illustrated in Fig. 2c), we disentangled the goal progress component from the tree-search value in the MRI analysis. To do this, we residualized all forms of value to goal progress and used goal progress as an independent regressor, allowing us to identify where goal progress is separately tracked in the brain. In addition, since the tree-search value of an option is an approximation of its 'time to completion', it is highly dependent on the size of the net across different blocks. To account for this, we also residualized the tree-search value to net size and included net size as a separate regressor. In other words, for each value component (current goal, best alternative, worst alternative), we removed the components related to goal progress and goal size, and added these components as unique regressors to examine separately. All regressors were *z*-scored at the level of individual runs before fitting the GLM. Extended Data Fig. 6a displays the final correlations between the regressors.

In addition to the parametric regressors, five types of events were included in the final GLM as main effects: onset of the decision period, onset of the block, spatial presentation of the three stimuli (substituting the spatial task), the update of the net and the end of the block. The following confound regressors were also included in the design matrix: six motion regressors produced during realignment, the physiological explanatory variables (processed by PNM) and a matrix of motion outlier timepoints. Motion outliers were detected using FEAT's fsl_motion_outliers tool. Metric values for detecting motion outliers were calculated for each timepoint using the root mean square intensity difference between each volume and the reference volume, and outliers were identified as volumes for which the metric value exceeded the 75th percentile + 1.5 times the interquartile range. Note that no participants or timepoints were removed from analyses due to motion, but rather, the effect of these outlier timepoints on the analysis was controlled for by including a confound matrix of outlier timepoints in the GLM. Across runs, the median percentage of timepoints identified as outliers was 2.4% of volumes (maximum across all runs 9.1%).

*Whole-brain intertrial analysis.* Given that behavioural biases accompanying goal pursuit lasted even outside of the decision period (in our spatial task), we asked whether goal-related neural activity also persisted between decisions. Of the regressors listed under 'Univariate fMRI analyses', goal progress is the one dynamic variable that can be tracked between trials (rather than depending on information presented at the decision; that is, the offers which feed into the option values). We therefore specifically investigated whether information about goal progress was carried between trials.

To do this, we ran a whole-brain analysis where we included all the same regressors listed in 'Decision-time analysis', both time-locked to the decision onset and time-locked to the presentation of the first fixation (ITI 1; see Extended Data Fig. 1d for ITI timing during task and Extended Data Fig. 6d for the regressor correlation matrix). We asked whether the activity tracking goal progress was present during the ITI (see Fig. 3b for results of this analysis and Extended Data Table 1 for the table of fMRI peak activity).

**ROI analyses.** *ROI selection.* The vmPFC, ventral striatum and dorsal ACC (dACC) were selected as regions-of-interest for further analysis because they showed strong value-related activity at decision time in our whole-brain analysis. This is consistent with previous literature showing that the dACC is involved in value-guided abandonment[27,29,31], and the ventral striatum is a centre of value-guided choice[63], known to be sensitive to goal proximity[45] and with meaningful projections to vmPFC[64]. Given the relevance of these areas for decision making during goal pursuit, we created regions of interest at the peaks of

activity in these areas from our whole-brain analysis. Full details of the extraction procedure for the ROIs used in this paper can be found in Supplementary Information.

*Baseline activity analysis.* As in previous paradigms[23], we defined baseline activity as the activity present at the onset of the choice offers, before the new offers or the decision itself influenced the dynamics (that is, $t = 0$ of the time course shown in Extended Data Fig. 7d–f). Full details of the baseline activity analysis are included in Supplementary Information.

*Value modulation analyses.* To determine how neural representations of value are modulated by goal pursuit, we investigated the interaction between goal progress and value. Following the behavioural analyses, this involved predicting neural activity using the interaction between goal progress and each source of value (tree-search model value of best alternative and tree-search model value of current goal). Full details of the analysis procedure within ROIs are included in Supplementary Information.

### Lesion patient study

**Participants and experimental procedure.** Twenty-six patients with brain lesions (13 female; mean age 58 years) and 27 age-matched control participants (17 female; mean age 59 years) took part in the study. Of the lesion patients, one was excluded because they failed to pass the initial comprehension questions and two were excluded because they were unable to complete the task. Of the remaining 23 individuals in the study, 16 had damage within the frontal cortex and the remaining 7 had damage to other areas (see Fig. 5a for maps of lesion overlap). No statistical methods were used to pre-determine sample sizes, but our sample sizes are larger than those reported in previous publications[41,43,44]. The patient population was recruited from a database of individuals who had previously visited the John Radcliffe Hospital and consented to be contacted for research studies. Ethical approval for the patient study was obtained from the London Fullham Research Ethics Committee (IRAS project number: 242551, REC Reference number: 18/LO/2152). This study was not pre-registered. All participants gave written informed consent before the experiment. Participants were paid £15 per hour plus a performance-dependent bonus of £8–12. Data collection took place online over a single session where the participant completed an online version of the task (hosted on Pavlovia), while the researcher remained on the telephone throughout the session. Before beginning the task, the participant received 12 trials of training and was required to pass three comprehension questions before proceeding to the main task, which consisted of 250 trials in total. The same schedule was used across all participants. The age-matched controls completed the same schedule and training procedure online, and were recruited through Prolific.co.

**Voxelwise lesion analysis.** We began by investigating the relationship between brain damage and persistence biases independently from the fMRI study. To investigate areas causally relevant for persistence in the task, we performed a voxelwise whole-brain analysis predicting behaviour from maps of the patients' neural damage (Fig. 5b). For each voxel, we predicted individual persistence biases using a binary regressor capturing whether the voxel was damaged in that individual:

$$\text{Persistence bias} \sim \text{Voxel damage} \qquad (6)$$

We used a threshold of $t > 2.3$, where damage predicted lower persistence biases ($P < 0.01$, one-sided test because our hypotheses concerned areas where damage leads to a 'reduction' in persistence biases).

*Permutation-based cluster correction.* We controlled for multiple comparisons by performing cluster correction using the false discovery rate

(FDR) method[65]. Using a permutation-based approach, we determined the maximum cluster size expected from our lesion dataset due to chance, using the same significance threshold. On each permutation (total of 1,000 iterations), we shuffled individual persistence biases and performed the same voxelwise regression analysis with the shuffled biases. We created a distribution of clusters found across all permutations and defined the minimum cluster size for significance at the 95% cut-off of all clusters found by chance, resulting in a minimum cluster size of 255 voxels.

**ROI-based lesion analysis.** Next, we performed a groupwise comparison where we split lesion patients on the basis of whether they were damaged in a region pre-defined by our fMRI study. Our fMRI study had identified a subset of areas carrying signals relating to goal pursuit even between decisions, focused on the vmPFC. We split all patients into two groups on the basis of whether they were damaged at an ROI centred on the peak of this interdecision fMRI activity. Following the same procedure described in ROI selection and extraction procedure, we extracted a region of interest with a 3-voxel radius (7.2 mm$^3$) centred on the peak of activity tracking goal progress during the ITI in our fMRI study. We then tested for a difference in persistence biases between the two groups of patients and against the age-matched controls. We used a one-sided permutation test to test for a difference in means between groups due to the small sample sizes and non-normally distributed biases (Fig. 5c; we used a one-sided test based on our hypothesis that damage to the vmPFC would reduce persistence, although we note that the difference remains significant if we were to perform a two-sided test).

**Patient control analyses.** Our voxelwise regression analysis identified a region of the vmPFC that included damaged voxels from five different patients. Our ROI-based lesion analysis independently identified four out of the five same patients when selecting on the basis of a pre-defined fMRI region. For the subsequent control analyses, we verified that the initial five patients were truly less biased to persist, rather than persisting less for other reasons (such as using a drastically different strategy or responding more randomly). We note that if these control analyses were limited to the four patients identified in the ROI-based analysis (excluding the additional vmPFC patient identified in the voxelwise analysis), the same conclusions hold.

First, we compared performance across groups. If vmPFC patients are truly less 'biased' to persist than other patients, rather than just being more random in their switch behaviour, we should expect to see a performance enhancement. We quantified performance as the mean number of trials taken to complete a goal, where a lower value means goals were completed faster. Since all participants in the patient task completed the identical schedule, this measure is not vulnerable to schedule-specific artefacts. We then tested whether vmPFC patients performed better than patients with damage elsewhere, using a one-sided parametric test (Fig. 5d; we used a one-sided test based on our hypothesis that reduced bias should improve performance, but also note that the difference remains significant if we were to perform a two-sided test).

Second, we confirmed that behaviour among patients with damage to this region was still best explained by the same behavioural model as healthy individuals (the 'tree-search model') and not by a simpler strategy, by fitting the four behavioural models described in 'Model fitting' (Extended Data Fig. 9a).

Finally, we verified that the vmPFC patients were not more stochastic in their decision process. We quantified stochasticity as inverse temperature, which is the beta weight associated with the tree-search value in our logistic regression predicting abandonment choices. We used two-tailed permutation tests to verify that there was no difference in stochasticity between the vmPFC lesion group and other patients, and between the vmPFC lesion group and age-matched controls

(Extended Data Fig. 9b). Further information about performance in the spatial task among patients can be found in Supplementary Information.

## Reporting summary

Further information on research design is available in the Nature Portfolio Reporting Summary linked to this article.

## Data availability

All behavioral data and the preprocessed fMRI data have been deposited at OSF (https://osf.io/mvquk/) and are publicly available as of the publication date. The patient lesion maps are not publicly available as this would compromise the privacy of the research participants.

## Code availability

All analysis code is available on OSF at https://osf.io/mvquk/ (Identifier: DOI 10.17605/OSF.IO/MVQUK).

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

## Acknowledgements

This research was funded in part by the Wellcome Trust (Grant number 222347/Z/21/Z to E.H.). For the purpose of Open Access, we have applied a CC BY public copyright licence to any author-accepted manuscript version arising from this submission. This research was also funded in part by the Biotechnology and Biological Sciences Research Council (grant BB/ R010803/1 to N.K., https://bbsrc.ukri.org/) and the European Union (ERC to N.K., FORAGINGCORTEX, project number 101076247). Views and opinions expressed are, however, those of the authors only and do not necessarily reflect those of the European Union or the European Research Council Executive Agency. Neither the European Union nor the granting authority can be held responsible for them. J.G. was funded by the Medical Research Council UK (MR/K501256/1 and MR/N013468/1) and St John's College, Oxford. J.X.O. was funded by the Medical Research Council UK (MR/L019639/1). The patient study was also supported by an Oxford NIHR BRC, the McDonnell foundation and an MRC clinician scientist fellowship (MR/P00878X to S.G.M). We thank the Wellcome Centre for Integrative Neuroimaging, Oxford, for Magnetic Resonance Research, supported by core funding from the Wellcome Trust (203139/Z/16/Z and 203139/A/16/Z). The funders had no role in study design, data collection and analysis, decision to publish or preparation of the paper. We thank J. Scholl, M. Klein-Flugge, C. Summerfield and R. Holton for helpful discussions of this work.

## Author contributions

E.H., N.K., J.X.O. and J.G. contributed to conceptualization and methodology. E.H. contributed to software, validation, formal analysis, investigation and writing of the original draft. N.K., J.G. and J.X.O. contributed to supervision, and reviewing and editing of the paper. H.W. contributed to investigation in the lesion patient study. S.G.M. contributed to supervision in the lesion patient study, and reviewing and editing of the paper.

## Competing interests

The authors declare no competing interests.

## Additional information

**Extended data** is available for this paper at https://doi.org/10.1038/s41562-024-01844-5.

**Correspondence and requests for materials** should be addressed to Eleanor Holton.

# Article

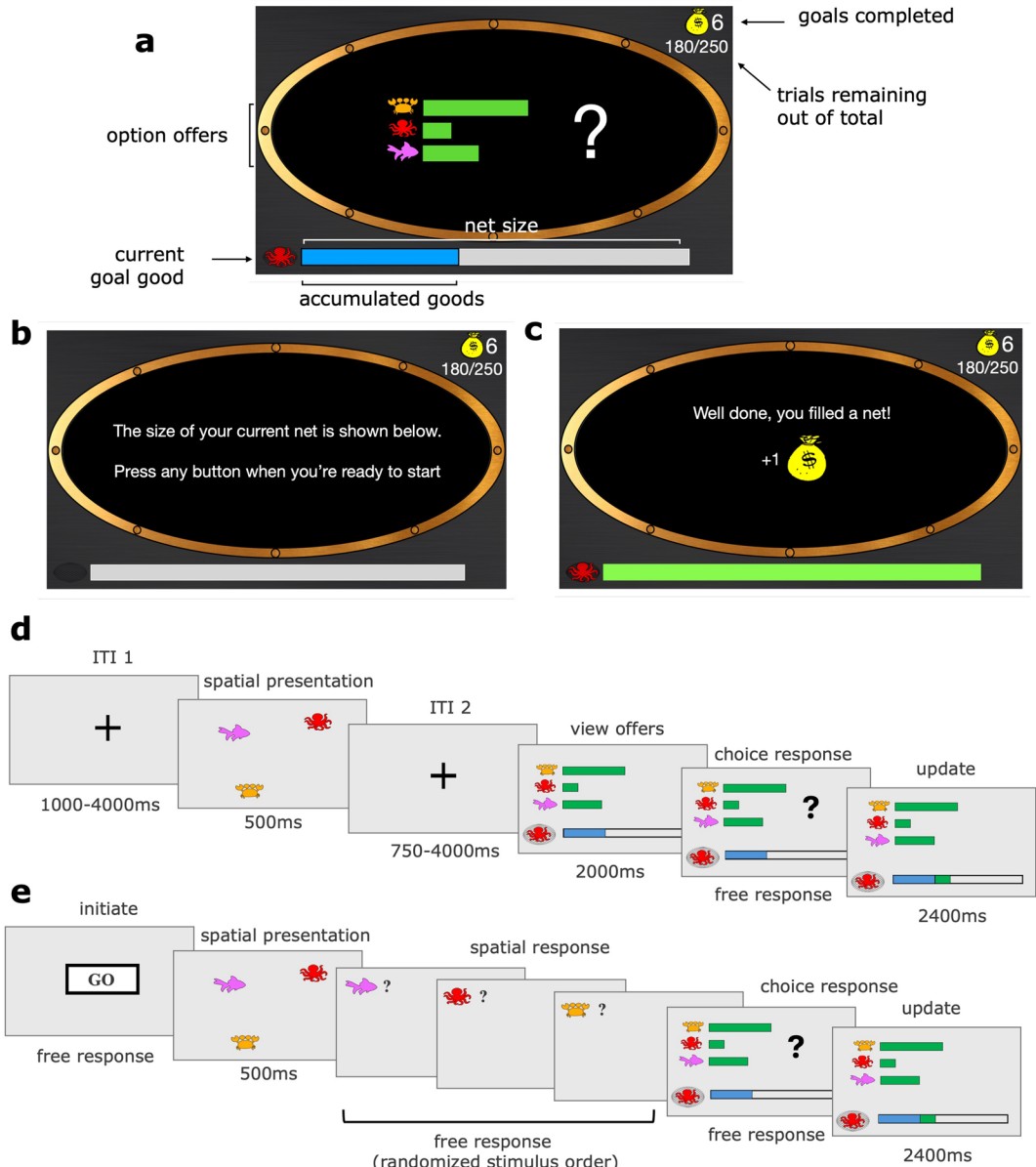

**Extended Data Fig. 1 | Task Design. (a)** Illustration of task as presented to participants using the framing story of an 'underwater' fishing game. In addition to the features described in Fig. 1a, the number of nets completed (that is points) was shown in the top right corner next to the money bag icon. Total nets were directly translated to the participant's bonus payment at the end of the study. The number of trials remaining in the session was shown directly underneath, as a proportion of the total trials in the session. This was to incentivise participants to make as strategic choices as possible to maximise the number of nets they could fill within the remaining trials. **(b)** At the start of each new block, participants were presented with the size of the net they needed to fill. **(c)** A block ended when the participant had filled the net, and a point was won. **(d)** Task sequence inside the scanner. To keep the task visually consistent with the spatial session outside

the scanner, participants passively viewed the three sea creatures flash on screen during the inter-trial interval, but were not required to report the location of the creatures. To dissociate activity related to the decision from activity related to response indication, we included a two second buffer zone once the offers were presented, before participants could make their response. In the main fMRI analyses, activity was time-locked to the onset of the decision period, shown here as 'view offers'. In the additional ITI analysis, activity was time-locked to 'ITI1'. **(e)** Task sequence in the spatial session (outside the scanner). Participants initiated the presentation of the creatures. After viewing the presentation of the creatures for 500 ms, they were then probed on the location of the three creatures in a randomised order before being presented with the main decision task.

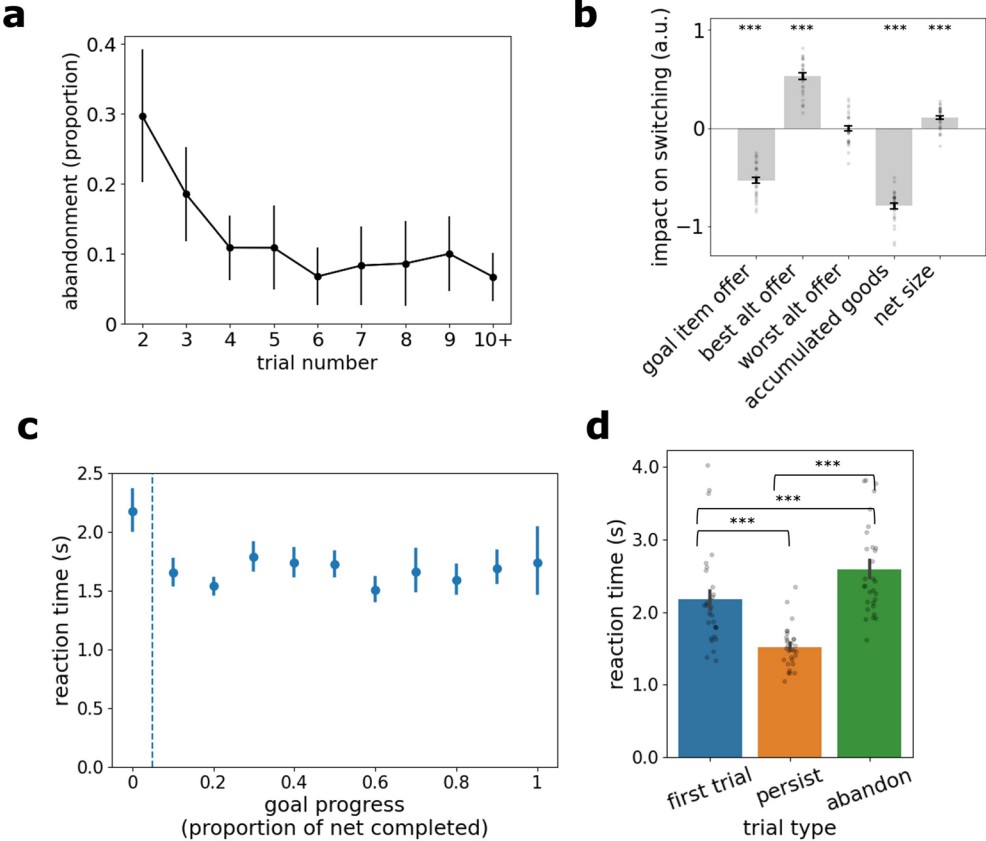

**Extended Data Fig. 2 | Task behaviour. (a)** Proportion of abandonment choices as a function of trial number (mean values +/− SD, $n = 30$). **(b)** Results of a logistic regression analysis predicting goal switching as a function of simple task parameters (mean beta weights associated with the regressors are plotted, +/− SEM, $n = 30$, dots show individual participant beta weights). The plot demonstrates participants were sensitive to the key elements of the task: they were more likely to switch when the offers associated with alternative goods were high (two-sided t-test against zero; $t(29) = 14.74, p = 5.29 \times 10^{-15}$), and less likely to switch when the offers for their current good were high (two-sided t-test; $t(29) = −16.84, p = 1.63 \times 10^{-16}$), or after having accumulated many goods in their net (two-sided t-test; $t(29) = −27.67, p = 2.12 \times 10^{-22}$). We also found that people were more likely to abandon a goal when the size of the target net was larger (two-sided t-test; $t(29) = 5.35, p = 9.51 \times 10^{-6}$). We found no effect of the second-best alternative on abandonment decisions ($t(29) = 0.11, p = 0.912$). **(c)** Reaction times plotted against goal progress (mean values +/−SEM, $n = 30$). **(d)** Relationship between reaction times and trial type (mean values +/− SEM, $n = 30$, dots show individual participant means). Participants are much slower to respond on abandonment trials compared to persist trials (mean RT on persist trial = 1.52 seconds, $SD = 0.28$; mean RT on abandonment trial = 2.59 seconds, $SD = 0.62$; two-sided paired t-test for difference in RTs: $t(29) = 11.05$, p $= 6.42 \times 10^{-12}$). Participants were also faster on persist trials than the first trial of a block (mean RT on the first trial = 2.17 seconds, $SD = 0.65$; two-sided paired t-test for difference between first trial and persist trial: t(29) = 6.05, $p = 1.38 \times 10^{-6}$), but slower on abandonment trials than the first trial (two-sided paired t-test for difference between first trial and abandon trial $t(29) = 2.99$, p = 0.006).

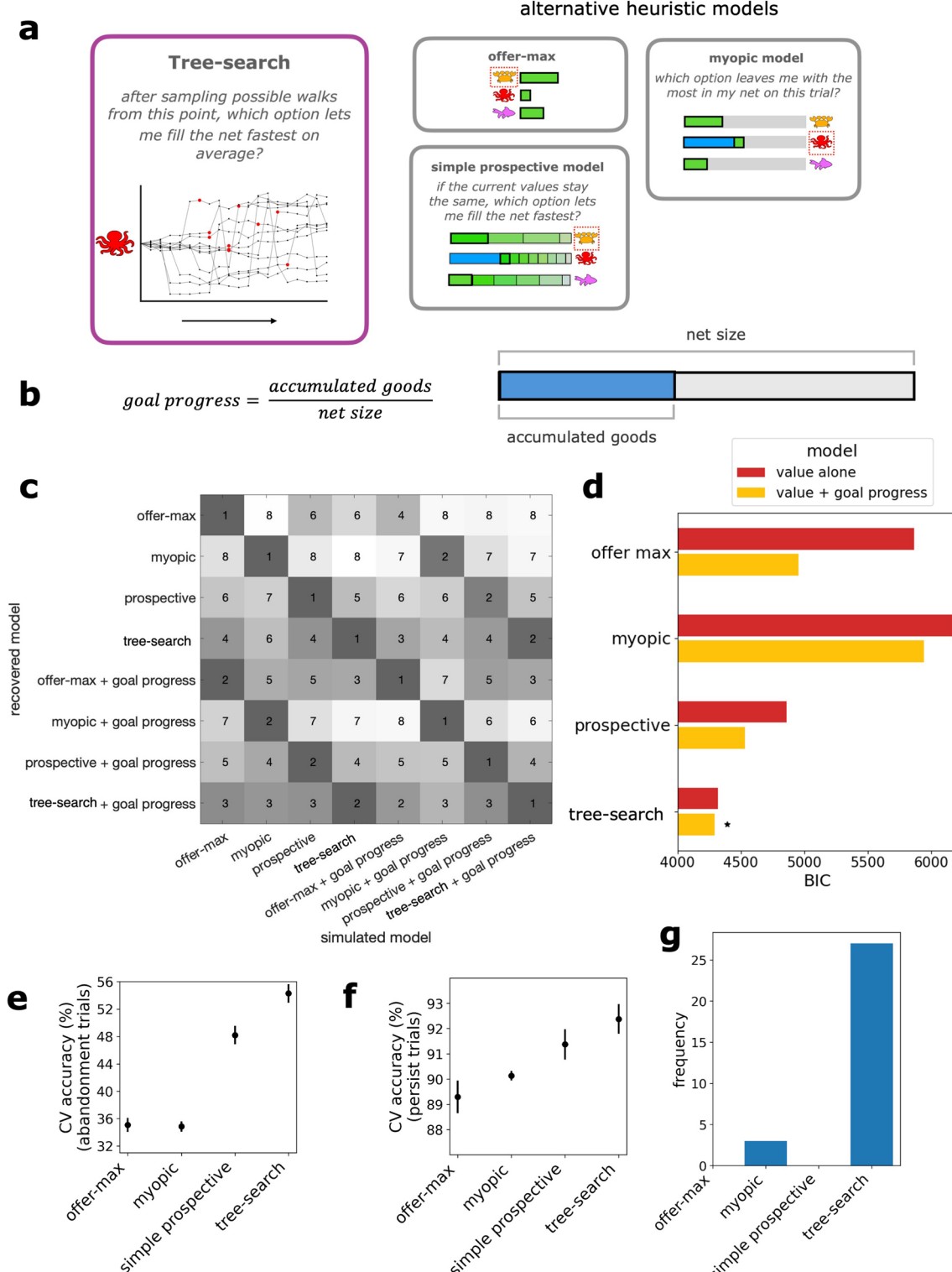

**Extended Data Fig. 3 | See next page for caption.**

**Extended Data Fig. 3 | Behavioural models. (a)** Graphic of the four behavioural models as described in 'Behavioural Models' in methods. **(b)** Graphic of the 'goal progress' regressor, defined as the proportion of the goal completed (net contents / net size). Note that goal progress differs from overall progress in the study, since multiple goals (nets) are completed successively over the course of the study. **(c)** Confusion matrix resulting from model recovery procedure. Each column corresponds to a model used to simulate the dataset. Each row corresponds to the model used to recover the dataset. Within a column, shading corresponds to the BIC of each competing model relative to the winning model. Lower BICs corresponding to better fits are displayed in darker shades. Numbers indicate the rank of the model in the model comparison per column (where 1 is the winning model, and 8 is the worst fitting model). In all cases, simulated behaviour is best fit by the true generative model. **(d)** Behavioural information criteria (BICs) for the models fitted to participant behaviour. Each model fitted corresponds to a model recovered in (c). Dark orange depicts logistic regression

models using model value alone. Light orange depicts logistic regression models with goal progress added as an additional regressor. **(e)** Cross-validation accuracy of each model predicting only abandonment trials. A leave-one-out procedure was used. For each participant, we fit each of the mixed-effects model to the choices of all other participants ($n = 29$). Predictive accuracy was computed from the fixed effects on the left-out participant. Mean cross-validated performance across participants is plotted, with error bars depicting SEM. **(f)** Cross-validation accuracy of each model predicting only persistence trials. Mean cross-validated performance across participants is plotted, with error bars depicting SEM. As for (e), the tree-search model describes behaviour best. **(g)** Best-fitting model frequencies across the fMRI healthy population. For each participant, the best fitting model was assessed using the cross-validated accuracies. The tree-search model was the best fit to choices for 27 out of 30 participants.

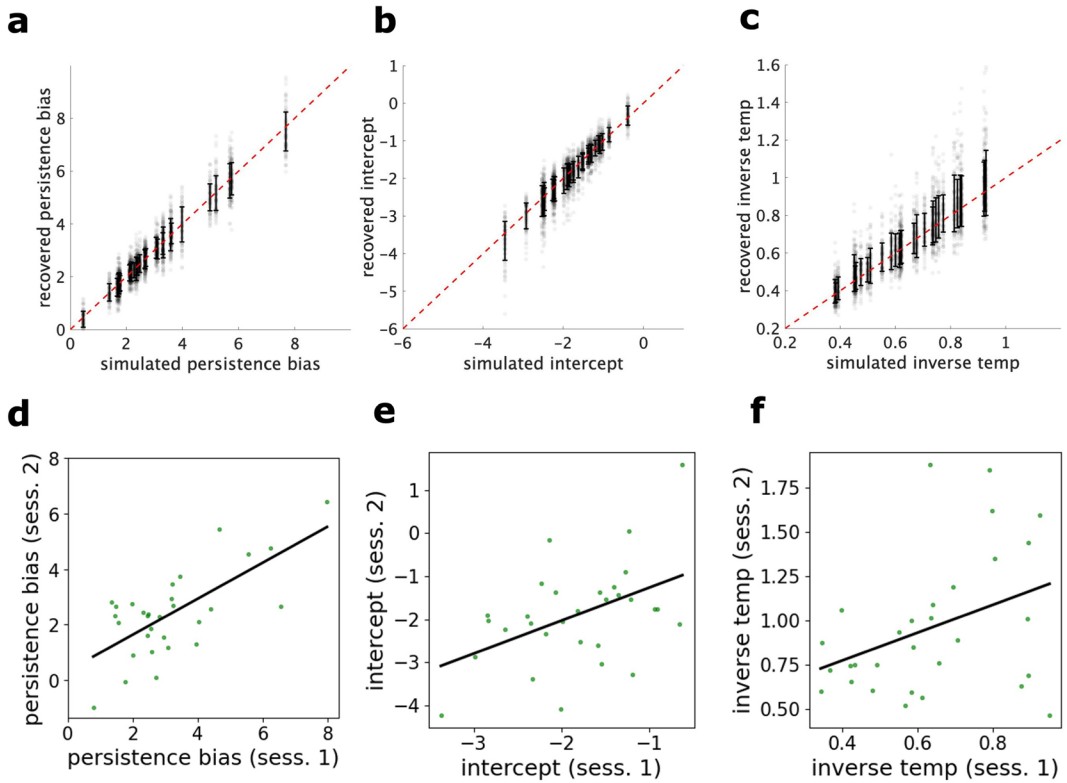

**Extended Data Fig. 4 | Parameter recoveries and test-retest reliability.**
**(a, b, c) Parameter recoveries**. Empirical parameters from decision data
aggregated across both sessions were used to simulate behaviour (100 iterations
per participant yielding $n = 300$ simulations total). Parameters were recovered
for each simulation by fitting a logistic regression using the same procedure used
for the empirical data. Mean recovered parameter +/− SD are plotted, with dots
showing recovered parameters for individual iterations. Red dotted line indicates
the identity line (perfect recovery). Two-sided Pearson's correlations were
performed. **(a)** An individual's persistence bias was defined as their 'indifference
point' to abandonment when predicting abandonment choices using the
tree-search value of abandonment. This is equal to the negative product of
temperature and intercept from the logistic regression (-intercept*temperature;
see 'Persistence bias' section in Methods). Recovered persistence biases
correlated with the simulated biases with a Pearson's correlation of 0.96
($p = 6.74 \times 10^{-170}$). **(b)** Recovery of the intercept parameter: Recovered against

simulated parameters from the logistic regression. The simulated intercepts can
be recovered with a Pearson's correlation of 0.92 ($p = 3.44 \times 10^{-122}$).
**(c)** Recovery of the inverse temperature parameter: Recovered against simulated
inverse temperature. The simulated inverse temperature can be recovered with
a Pearson's correlation of 0.84 ($p = 2.46 \times 10^{-82}$). **(d, e, f) Test-retest reliability of
parameters across the two sessions**. Parameters were separately fitted to the
decision task inside the scanner ('session 1', see Extended Data Fig. 1d) and to the
decision task outside the scanner ('session 2', Extended Data Fig. 1e). Two-sided
Pearson's correlations are reported. All parameters show significant test-retest
reliability. **(d)** Test-retest reliability for persistence biases across the two sessions
(Two-sided Pearson's $r = 0.69$, $p = 2.48 \times 10^{-5}$). Persistence biases are derived from
the intercept and inverse temperature shown in e and f. **(e)** Test-retest reliability
for the intercept alone across the two sessions (Pearson's $r = 0.46$, $p = 0.010$)
**(f)** Test-retest reliability of inverse temperature across the two sessions
(Pearson's $r = 0.38$, $p = 0.040$).

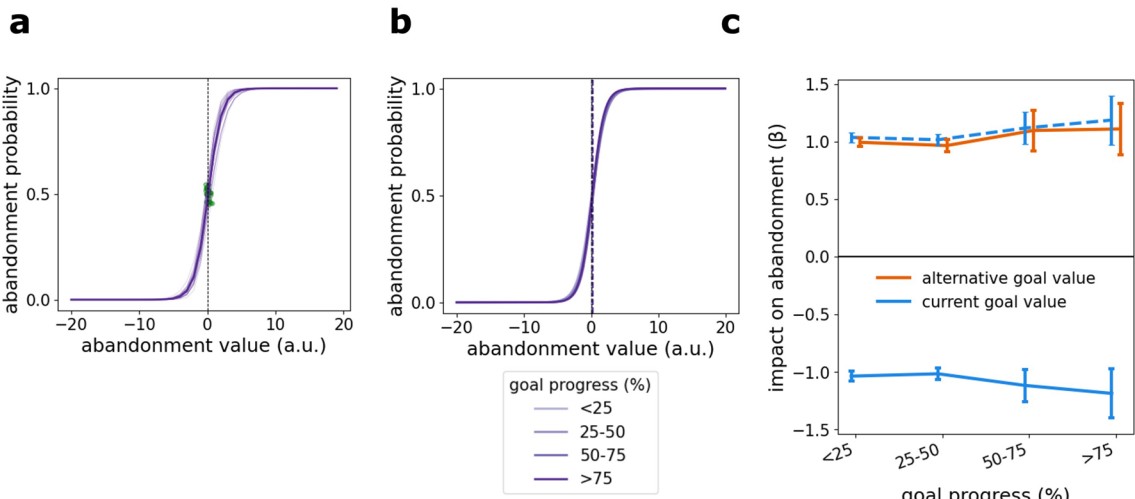

**Extended Data Fig. 5 | Simulated behaviour from tree-search model.**
Behavioural analyses on simulated data from the normative tree-search model, demonstrating the empirical behavioural patterns are not an artefact of the testing schedules. Here, we fit the normative tree-search model to 30 simulated data-sets in which each simulation corresponded to one of the 30 participant schedules. **(a)** Individual fits to simulated data sets show persistence biases of zero, demonstrating that the empirically found biases are not an inherent feature of the schedules used. Compare to Fig. 2b showing the range of participant specific persistence biases. **(b)** Quartile fits demonstrating that we accurately recover no difference in persistence biases across quartiles when simulating with the normative model. Compare to Fig. 2c showing that participants are more biased to persist as goal progress increases. **(c)** Effects of temptation and frustration on abandonment choices. In the normative model simulations, we accurately recover no difference in how these two abandonment causes are weighted. Compare to participant behaviour in Fig. 2d showing diverging slopes corresponding to the impact of frustration vs temptation across goal progress.

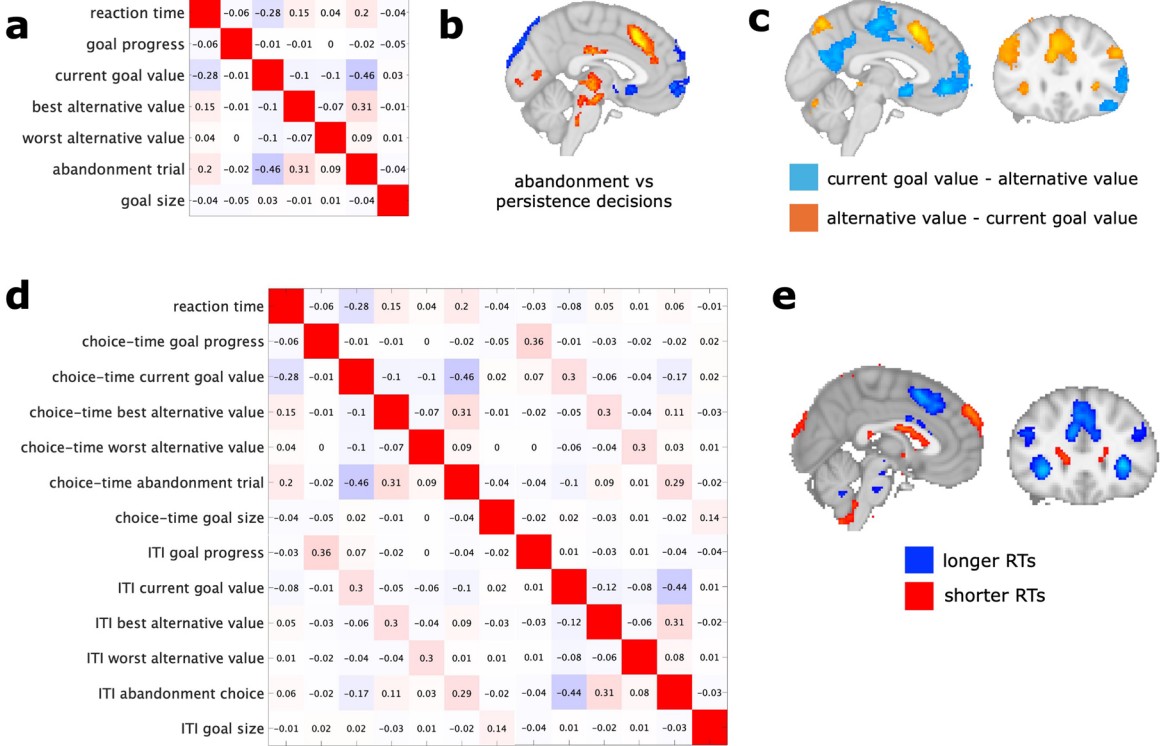

**Extended Data Fig. 6 | Whole-brain fMRI analyses. (a)** Correlation matrix (convolved with the hemodynamic response function) of the regressors included in the initial whole-brain analysis, alongside event regressors. The regressors were time-locked to the onset of the choice (that is 'view offers' in Fig. 1d). Value refers to value from the tree-search model, the normative model which described behaviour best. Value was orthogonalised to goal progress to separate these components each shown to affect behaviour (see Methods). **(b)** Activity related to abandonment (red) vs persistence (blue) decisions. In other words, the positive and negative activity related to the 'abandonment trial' regressor in (a), where trials were coded as 1 for abandonment and −1 for persistence **(c)** Activity related to the value contrasts. Blue depicts areas where activity increases when the tree-search model value of persisting with the goal is greater than the tree-search value of switching to the best alternative. Orange shows the inverse

(where activity increases when the tree-search model value of switching to the best alternative is greater than the tree-search model value of persisting with the current goal). The peaks of these value contrasts were used to define the regions of interest in vmPFC, ventral striatum, and ACC, as shown in Extended Data Fig. 7a–c. **(d)** Correlation matrix (convolved with the hemodynamic response function) of the regressors included in the second ITI whole-brain analysis. This analysis included all the same regressors as the main whole-brain analysis, this time modelling BOLD activity both at the choice onset, and in addition at the first inter-trial fixation cross (that is 'ITI 1' in Fig. 1d). **(e)** Relationship between log reaction times and whole-brain activity, at the time of decision. The areas where longer RTs predict more activity include ACC, insula, and dlPFC. The areas where shorter RTs predict stronger activity include ventral striatum and frontal pole (area 9). We do not see activity related to RTs in vmPFC.

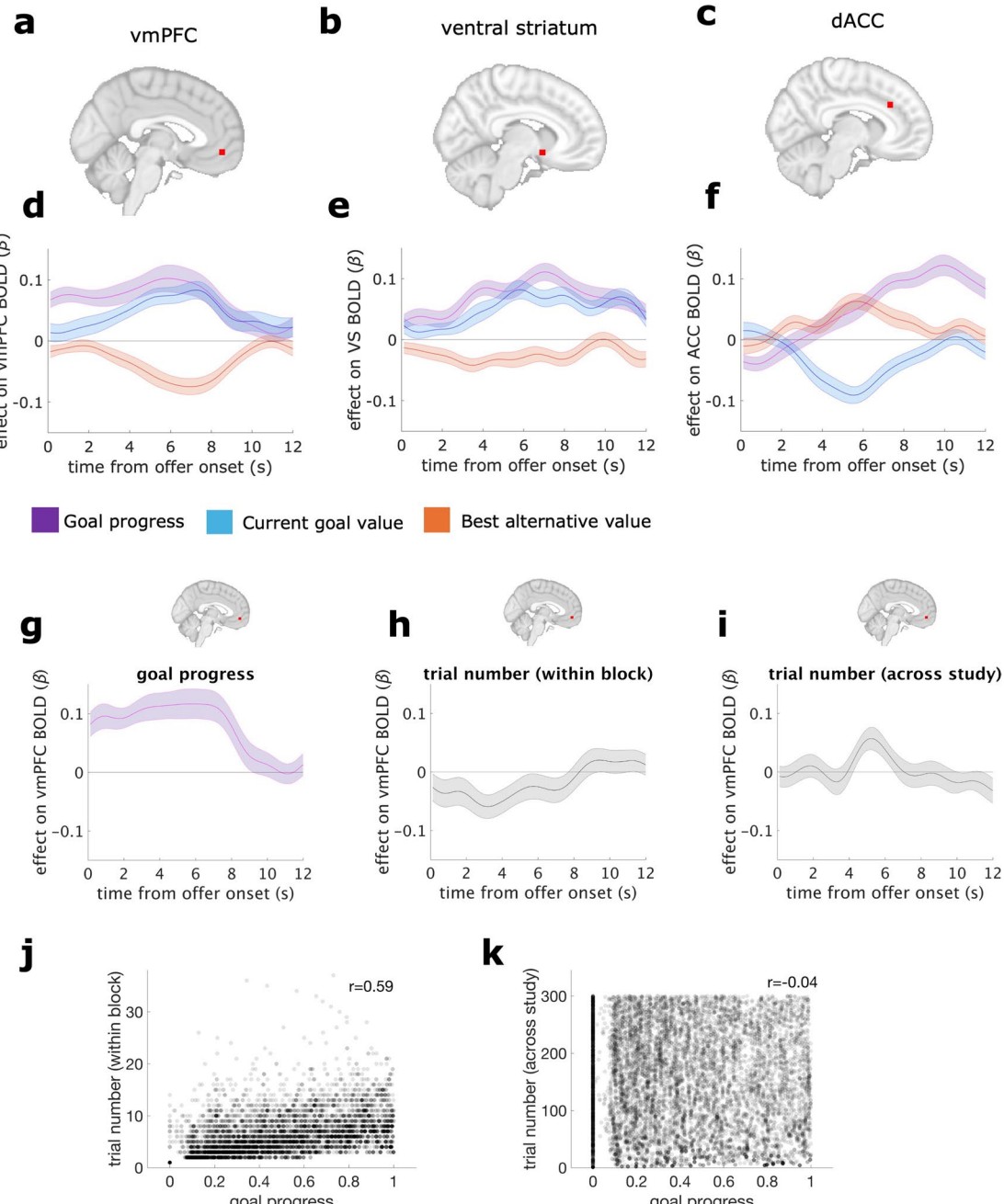

**Extended Data Fig. 7 | Regions of interest and neural activity time courses.**
**(a, b, c)** We extracted regions of interest based on the peaks of value-related activity in our fMRI study (see '*ROI selection and extraction procedure*' in Methods). These consisted of the peaks of activity for the contrast of goal value–alternative value in the case of vmPFC [−2,48,−8] and ventral striatum [8,8,−10], and the largest sub-peak of activity in the dACC for the alternative value–goal value regressors [8,28,30]. All peaks are shown in Extended Data Table 1 (ROI peaks starred). **(d, e, f)** Time course analyses depicting the t-statistics for the regressors of goal progress (purple), current goal value (blue), and best alternative value (orange) in the three regions of interest (for illustration). Time 0 seconds corresponds to the onset of the choice (that is 'view offers' in Extended Data Fig. 1d). The GLM used in the time-course analysis contained identical regressors to the whole-brain GLM described in *Univariate fMRI analyses* in

Methods, and shown in the correlation matrix in Extended Data Fig. 6a. Mean beta weights are plotted, where shaded error show SEM across participants (n = 30). Note the pre-decision modulation of activity by goal progress (t = 0) in the vmPFC predicted individual differences in attention and decision (Fig. 3d, e). **(g, h, i)** Additional analysis controlling for possible confounds of goal progress within the same GLM. Mean beta weights are plotted, where shaded interval shows SEM (n = 30). **(g)** Even when controlling for both within-block and across-study trial number, vmPFC baseline activity tracks goal progress. **(h)** VmPFC activity does not track within-block trial number at baseline. **(i)** VmPFC activity does not track trial number across the study. **(j, k)** Goal progress (that is proportion of the goal completed) is correlated with trial number within a block (**j**; where a block corresponds to a single goal), but is not correlated with total trial number across the study (**k**).

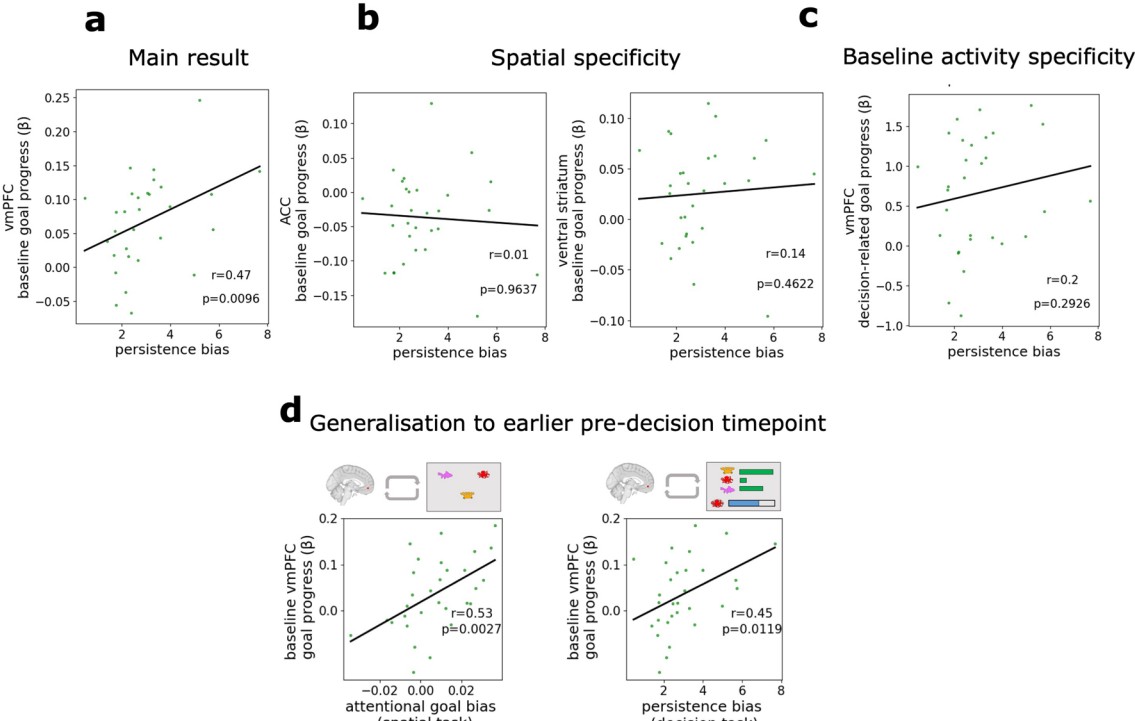

**Extended Data Fig. 8 | Control analyses for the relationship between baseline vmPFC activity and persistence bias.** Statistics shown in each plot report two-sided Spearman's correlations (effect and significance) between the neural regressor, and the persistence bias fitted to data aggregated across both sessions. In all plots, green dots show individual participant biases plotted against the relevant neural regressor beta weight. **(a)** Main effect from manuscript, showing baseline vmPFC tracking of goal progress predicts individual differences in persistence bias (where baseline representation of goal progress is defined as the beta weight for the relationship between vmPFC BOLD activity and goal progress at decision onset). Our decision to examine baseline vmPFC activity was based on hypotheses from previous literature showing baseline vmPFC activity carries subjective biases in decision-making, but here we present various controls. **(b)** Control analysis showing spatial specificity of the effect in (a) to vmPFC: Baseline representation of goal progress in the ACC and striatal regions of interest does not predict individual differences in persistence biases. **(c)** Decision-related representations of goal progress in vmPFC does not

predict individual difference in persistence bias. Here, we capture decision-related representation of goal progress by multiplying the fitted beta coefficients for goal progress at each time-point from choice onset by the double gamma HRF function, and summing the products to produce a decision related component for each participant (same procedure described in *Value modulation analyses*). **(d)** In the main manuscript, baseline vmPFC was defined as unconvolved activity at the moment (t = 0) of choice-onset (as in Vinckier et al. 2018.) Here we show the relationship between baseline vmPFC and behaviour is unaffected if baseline activity is defined as the unconvolved activity two seconds prior to choice-onset (as in Lopez-Persem et al. 2016). In other words, to address concerns that activity could be related to the onset of the choices, here we show that activity at a time-point two seconds earlier is predictive of behavioural biases, consistent with the interpretation that pre-decision activity is the critical predictor. In contrast, **(c)** shows that decision-related activity (post choice onset), is not predictive of individual differences.

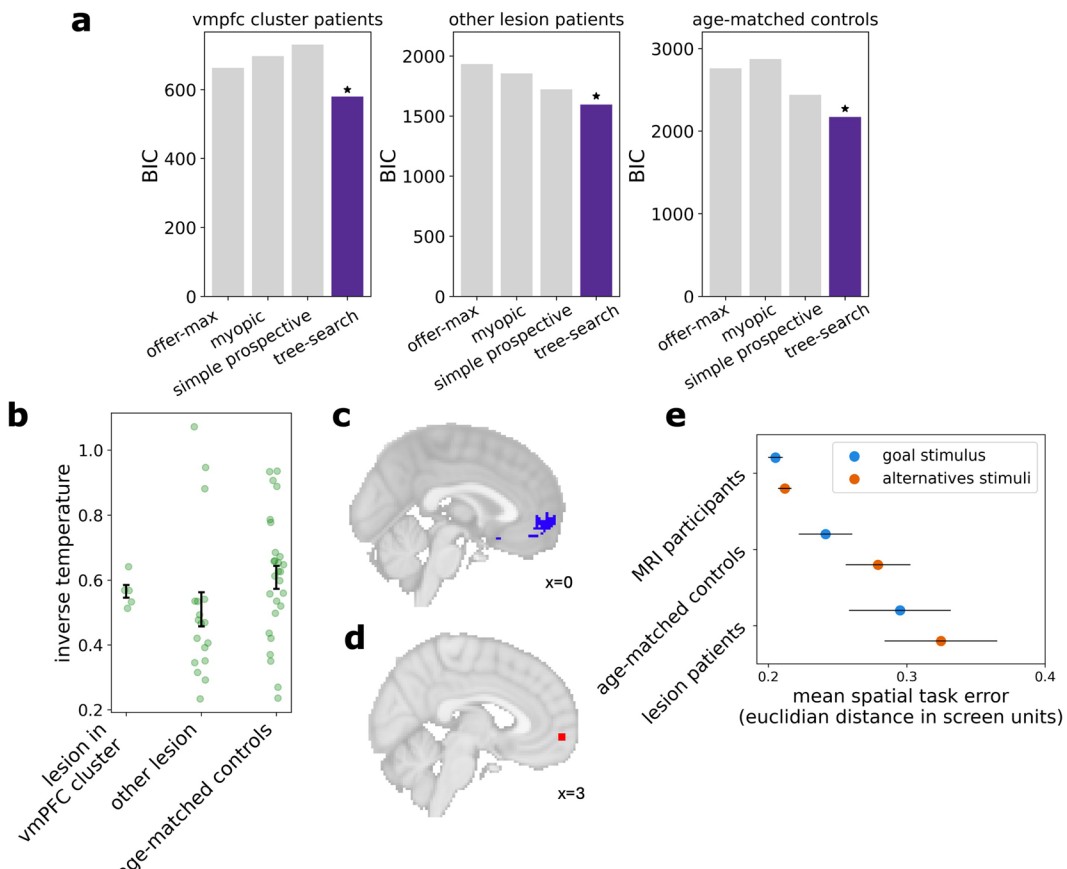

**Extended Data Fig. 9 | Lesion patient additional analyses. (a)** Results from a post-hoc control analysis where we fit the four normative models to behaviour in the three groups (left: patients with damage within the vmPFC cluster identified from the voxel-wise analysis; middle: patients with damage external to the vmPFC cluster; right: age-matched healthy control participants). In each group, we fit the models using a mixed effects regression model predicting abandonment choices (same method as for fMRI participants, described in Methods). Our results show the tree-search model remains the best fit to behaviour in all three groups. **(b)** Post-hoc analysis comparing inverse temperature across the three groups described in (**a**). Patients with damage within the vmPFC cluster identified in the voxel-wise analysis are not simply more stochastic since they show no difference in inverse temperature to the age-matched controls (Inverse temperature in vmPFC patients: $n = 5$, mean = 0.57, std = 0.04; inverse temperature in other patients: $n = 18$, mean = 0.51, std = 0.22; inverse temperature in age-matched controls: $n = 27$, mean = 0.61, std = 0.19; difference between vmPFC group and other patients: two-sided permutation test, difference in means = 0.06, $p = 0.572$,

*n.s.*; difference between vmPFC patients and age-matched controls: two-sided permutation test, difference in means = 0.04, $p = 0.633$, *n.s.*). Note the inverse temperature parameter has high parameter recoverability (Extended Data Fig. 4c). **(c)** Voxel-wise map of t-statistics where lesion damage predicts reduced persistence biases (same as shown in Fig. 5b thresholded at a higher significance level for illustration, t > 2.6). **(d)** Region of interest from fMRI study, taken at the peak of activity tracking goal progress between decisions, at [4,58,−6] (see Extended Data Table 1). Patients are split into those with damage inside this ROI and those with damage external to this ROI in Fig. 5c. **(e)** Response errors in the spatial task for each group, where error is mean Euclidian distance between participant response and true item locations, in normalised screen units. Mean error +/− SEM is plotted ($n = 23$ patients), where orange shows error for the current goal stimulus, and blue shows error for the alternative goal stimuli. While the spatial bias effect is in the right direction, the patient group do not show a significant accuracy advantage for the current goal item relative to alternative goal items.

**Extended Data Table 1 | Peaks of activity from cluster-corrected whole-brain fMRI analyses**

| Contrast | Region | Peak coordinates (x,y,z in mm MNI space) | Z Value |
|---|---|---|---|
| **Whole-brain fMRI analysis time-locked to decision onset.** | | | |
| **Persistence value**: current goal value–best alternative. value | Ventromedial prefrontal cortex (area 11m) | -2, 48, -8* | 5.2 |
| | Ventral striatum | 8, 8, -10* | 5.16 |
| | Lateral frontal pole | -38, 30, -16 | 5.26 |
| | Supplementary motor cortex | 0, -6, 58 | 5.8 |
| | Parietal operculum cortex | Left:- 58, -38, 26; Right: 54, -28, 24 | Left: 5.98; Right: 5.62 |
| | Intracalcarine cortex | 8, -84, 4 | 6.94 |
| | Precuneous cortex | -6, -54, 12 | 5.55 |
| | Superior parietal lobule | 28, -40, 64 | 5.04 |
| | Parahippocampal gyrus | -24, -36, -14 | 4.8 |
| | Precentral gyrus | Left: -60, 0, 30 ; Right: 42, -18, 62; | Left: 4.69; Right: 4.6 |
| | Cerebellum area VIIa | 14, -64, -46 | 4.36 |
| | Cerebral crus | 26, -80, -32 | 4.56 |
| **Abandonment value**: best alternative value–current goal value | Dorsolateral prefrontal cortex | Left: -46, 44, 16; Right: 48, 36, 26 | Left: 5.74; Right: 6.27 |
| | preSMA extending into dorsal ACC | -4, 20, 46 dACC subpeak: 8, 28, 30* | 6.27 dACC subpeak: 5.4 |
| | Insular cortex | Left: -32, 20, 4; Right: 36, 18, 6 | Left: 5.77; Right: 5.49 |
| | Lateral frontal pole | 42, 44, 0 | 4.41 |
| | Inferior frontal gyrus / precentral gyrus | -44, 8, -32 | 5.74; |
| | Supramarginal gyrus | 50, -48, 48 | 6.21 |
| | Superior frontal gyrus | -26, 2, 62 | 4.76 |
| | Intracalcarine cortex | -8, -74, 8 | 6.03 |
| | Cerebellum area VI | -8, -74, -22 | 5.53 |
| | Cerebellum area VIIb | -34, -68, -56 | 4.93 |
| **Persistence choice**: Persistence trials–abandonment trials | Area 11m / Frontal medial pole | 0, 64, -8 | 4.95 |
| | Area 25 | 0, 20, -8 | 4.45 |
| | Orbitofrontal cortex (area 47) | -42, 28, -10 | 4.48 |
| | Occipital Pole | Left: -22,-104, -2; Right: 4, -88, 36 | Left: 4.3; Right: 4.6 |
| | Lingual gyrus | 10, -90, -18 | 4.49 |
| | Superior parietal lobule | -22, -46, 62 | 4.46 |
| | Central opercular cortex | Left: -56, 6, 0; Right: 62, 2, 8 | Left: 4.85; Right: 5.54 |
| | Precentral gyrus | 26, -22, 74 | 4.62 |
| | Lateral ventricle | Left: -20, -48, 16; Right: 36, -42, 4 | Left: 4.45; Right: 4.63 |
| | Cerebral crus | 18, -80, -40 | 3.95 |
| | Inferior temporal gyrus | 54, -58, -24 | 4.33 |
| | Lateral occipital cortex | -48, -76, 34 | 4.15 |
| **Abandonment choice:** Abandonment choice– persistence choice | Caudate | -12, 16, 4 | 5.57 |
| | preSMA extending into dorsal ACC | 6, 20, 44 | Right: 5.69 |
| | Dorsolateral frontal pole | -42, 42, 22 | 4.4 |
| | Lateral frontal pole (left) | -22, 62, 2 | 4.52 |
| | Posterior cingulate gyrus | -4, -20, 30 | 4.83 |
| | Insular cortex extending into frontal operculum cortex | Left: -32, 18, -6; Right: 36, 24, 4 | Left: 7.04; Right: 6.64 |
| | Occipital pole | -12, -94, -4 | 5.58 |
| | Superior parietal lobule | -38, -52, 50 | 4.41 |
| | Occipital fusiform gyrus | 24, -74, -4 | 4.81 |
| | Inferior frontal gyrus / precentral gyrus | -46, 2, 40 | 5.38 |
| | Superior frontal gyrus | -22, -2, 58 | 4.61 |
| | Lateral occipital cortex | -26, -84, 18 | 5.14 |
| **Goal progress** (i.e. proportion of net completed) | Non-region specific cluster (74958 voxels), encompassing areas stretching from lateral occipital cortex, temporal gyrus, insular cortex, striatum, cingulate cortex, and medial prefrontal areas | 10, -92, 6 | 8.1 |
| | Middle / superior frontal gyrus | -28, 34, 48 | 5.38 |
| | Brain stem | 2, -30, -42 | 4.6 |
| **Negative goal progress** (i.e. activity related to proportion of net *remaining* to be filled) | Lateral occipital cortex | -42, -64, 50 | 6.39 |
| Contrast | Region | Peak coordinates (x,y,z in mm MNI space) | Z Value |
| **Whole-brain analysis time-locked to the inter-trial fixation cross.** | | | |
| **Inter-trial goal progress** (i.e. goal progress time-locked to the ITI fixation cross) | Frontal medial pole (Area 10 stretching to area 11 and area 14) | 4, 58, -6* | 5.17 |
| | Right hippocampus | 26, -20, -14 | 4.69 |
| | Temporal pole | Left: -40, 22, -30; Right: 48, 18, -30 | Left: 4.34; Right: 4.47 |
| | Area 8m | -20, 36, 48 | 4.73 |
| | Precuneus cortex stretching to posterior cingulate gyrus | -6, -56, 22 | 4.48 |
| | Middle temporal gyrus | -64, -12, -14 | 4.46 |
| | Caudate | 14, 20, 6 | 4.08 |
| | Postcentral gyrus | 36, -24, 56 and second cluster at 54, -22, 54 | 4.45, 4.58 |

Family-wise error cluster corrected, z > 2.3, p < 0.05
*Coordinates used for region of interest analyses

Multiple comparisons were corrected for using a Z statistic threshold of 3.1, and a cluster probability threshold of p=0.05. See Extended Data Fig. 6 for all regressors included in GLM. Asterisks designate peaks of activity used to extract regions of interests.

| | |
|---|---|

# Reporting Summary

## Statistics

For all statistical analyses, confirm that the following items are present in the figure legend, table legend, main text, or Methods section.

| n/a | Confirmed | |
|---|---|---|
| ☐ | ☒ | The exact sample size (*n*) for each experimental group/condition, given as a discrete number and unit of measurement |
| ☐ | ☒ | A statement on whether measurements were taken from distinct samples or whether the same sample was measured repeatedly |
| ☐ | ☒ | The statistical test(s) used AND whether they are one- or two-sided<br>*Only common tests should be described solely by name; describe more complex techniques in the Methods section.* |
| ☐ | ☒ | A description of all covariates tested |
| ☐ | ☒ | A description of any assumptions or corrections, such as tests of normality and adjustment for multiple comparisons |
| ☐ | ☒ | A full description of the statistical parameters including central tendency (e.g. means) or other basic estimates (e.g. regression coefficient) AND variation (e.g. standard deviation) or associated estimates of uncertainty (e.g. confidence intervals) |
| ☐ | ☒ | For null hypothesis testing, the test statistic (e.g. *F*, *t*, *r*) with confidence intervals, effect sizes, degrees of freedom and *P* value noted<br>*Give P values as exact values whenever suitable.* |
| ☒ | ☐ | For Bayesian analysis, information on the choice of priors and Markov chain Monte Carlo settings |
| ☒ | ☐ | For hierarchical and complex designs, identification of the appropriate level for tests and full reporting of outcomes |
| ☐ | ☒ | Estimates of effect sizes (e.g. Cohen's *d*, Pearson's *r*), indicating how they were calculated |

*Our web collection on statistics for biologists contains articles on many of the points above.*

## Software and code

Policy information about availability of computer code

| | |
|---|---|
| Data collection | The experimental task used for data collection was custom developed by the authors using PsychoPy Standalone (Version v2021.1.2). |
| Data analysis | fMRI analyses were carried out using FSL, the FMRIB Software Library (FSL version 6.0.4). Behavioural and further neural analyses were carried out by custom-written scripts in python (3.11.5) and MATLAB (version R2021_a). Code for data analysis can be found in OSF at https://osf.io/mvquk/ (DOI Identifier: DOI 10.17605/OSF.IO/MVQUK.). The following python packages were used for data processing, analysis and visualization: pandas (2.1.1), numpy (1.26.0), seaborn (0.12.2), matplotlib (3.8.0), scipy (1.11.2), statsmodels (0.14.0), pingouin (0.5.3), rpy2 (3.5.11), nilearn (0.10.2), nibabel(5.1.0). |

For manuscripts utilizing custom algorithms or software that are central to the research but not yet described in published literature, software must be made available to editors and reviewers. We strongly encourage code deposition in a community repository (e.g. GitHub). See the Nature Portfolio guidelines for submitting code & software for further information.

# Data

Policy information about availability of data

All manuscripts must include a data availability statement. This statement should provide the following information, where applicable:

- Accession codes, unique identifiers, or web links for publicly available datasets
- A description of any restrictions on data availability
- For clinical datasets or third party data, please ensure that the statement adheres to our policy

> The behavioral data and preprocessed fMRI data has been deposited at OSF (https://osf.io/mvquk/) and is publicly available as of the publication date (DOI Identifier: DOI 10.17605/OSF.IO/MVQUK). The patient lesion maps are not publicly available as this would compromise the privacy of the research participants.

# Research involving human participants, their data, or biological material

Policy information about studies with human participants or human data. See also policy information about sex, gender (identity/presentation), and sexual orientation and race, ethnicity and racism.

| | |
|---|---|
| Reporting on sex and gender | We asked for participants' self-reported gender at the time of data collection: Out of a total of 30 fMRI participants, 19 self-reported as female. We do not include further analysis of gender, as it was not applicable to our research questions. |
| Reporting on race, ethnicity, or other socially relevant groupings | We did not collect data on race or ethnicity, as it was not applicable to our research questions. |
| Population characteristics | fMRI participants were majority University students (mean age of 25 years). Lesion patient participants were patients who had previously visited John Radcliffe Hospital Oxford and consented to being contacted (mean age of 58 years). Age-matched control participants were collected from an online recruitment platform, and lived in the UK (mean age of 59 years). All participants had normal or corrected-to-normal vision. |
| Recruitment | fMRI participants were recruited via email circulation on Oxford University mailing lists and Oxford-based social media platforms. Lesion patients were recruited by email after previously consenting to being contacted for research studies. Age-matched control participants were recruited online via the platform Prolific. There was no self-selection bias. |
| Ethics oversight | Ethical approval for the fMRI study was obtained by the Oxford Central University Research Ethics Committee (REC; Ref: R72921/RE001).<br>Ethical approval for the patient study was obtained by the London Fullham Research Ethics Committee (IRAS project number: 242551 REC Reference number: 18/LO/2152). |

Note that full information on the approval of the study protocol must also be provided in the manuscript.

# Field-specific reporting

Please select the one below that is the best fit for your research. If you are not sure, read the appropriate sections before making your selection.

☐ Life sciences ☒ Behavioural & social sciences ☐ Ecological, evolutionary & environmental sciences

For a reference copy of the document with all sections, see nature.com/documents/nr-reporting-summary-flat.pdf

# Behavioural & social sciences study design

All studies must disclose on these points even when the disclosure is negative.

| | |
|---|---|
| Study description | Data are quantitative experimental data. The fMRI study involves imaging data from a 50 minute fMRI scan, and behavioural data during the fMRI scan and in a 30 minute post-scan session. The lesion patient study includes online behavioural data from 23 lesion patients and 27 age-matched healthy controls using the same task paradigm as in the fMRI study. |
| Research sample | The research sample includes 30 healthy individuals (majority Oxford-based students) for the fMRI study, with mean age=25. This sample is representative of a healthy young population in the UK. For the lesion patient study, the sample included 23 brain-lesioned patients (mean age=58) and 27 age-matched control subjects (mean age=59). These samples are representative of a population of brain-lesioned patients and a population of healthy older individuals in the UK. Importantly, we do not compare behaviour between the younger fMRI group and the older lesion patient group since these samples are not matched. Instead we only compare lesion patient behaviour to the age-matched control population. |
| Sampling strategy | fMRI sample size (n=30) was chosen as the upper end of the recommended sample size for decision-making imaging studies, based in similar studies of naturalistic decision-making in fMRI paradigms (Juechems et al. 2019, Trudel et al. 2021, Park et al. 2021). Convenience sampling was used.<br>Lesion sample size (n=23) was limited due to patient availability, but is larger than those reported in similar lesion patient studies (Hare et al. 2011, Wolf et al. 2014, Noonan et al. 2017). This sample size is sufficient for recovering the effects we report, based on |

| | |
|---|---|
| | both previous analysis of healthy individuals, and the permutation based size-correction reported in the manuscript. Convenience sampling was used. |
| Data collection | fMRI study: training and post-scan tasks were undertaken on a laptop in a behavioural testing room, with one experimenter present. fMRI data were acquired at the Oxford Centre for Human Brain Activity using a 3T Siemens scanner. A trained radiographer was present in addition to the experimenter during the fMRI scanning.<br>Lesion patient study: both the lesion sample and the control sample performed the task virtually on a computer at their own home. For the lesion patients, the experimenter was remotely present during the study on the telephone.<br>In both cases, the researcher was not blind to the study hypothesis during data collection, but all training was standardised using computer-based task instructions and computer-based practice questions which were kept consistent across individual sessions. |
| Timing | Data collection for the fMRI study took place from May 2021 to January 2022 (with data collection limited by shortages in scanner availability at the end of the pandemic).<br>Data collection for the patient study was limited by lesion patient availability, with the first patient tested in May 2021 and the last patient tested in August 2022. |
| Data exclusions | One fMRI participant was excluded because they withdraw their participation in the study (before taking part in the scan itself). Two patients were excluded because they were unable to complete the task, and one patient was included because they were unable to pass the initial comprehension questions. |
| Non-participation | One participant dropped out of the fMRI study due to symptoms of claustrophobia in the scanner. Three lesion patients dropped out of the lesion patient study; one dropped out due to failing the initial comprehension test, and two dropped out early in the study due to fatigue. |
| Randomization | The experimental design of the fMRI study does not involve allocation of participants into different groups. Within the lesion study, data was analysed using two separate approaches. First, data from lesion patients was aggregated together (not allocated into separate groups) and analysed using voxel-wise regression alongside cluster correction methods (false discovery rate) to control for false positives. In the second analysis, lesion patients were allocated into two groups based on whether they were damaged within a region-of-interest pre-defined by the fMRI study. Potential group confounds such as task comprehension and age were controlled for during analyses. |

# Reporting for specific materials, systems and methods

We require information from authors about some types of materials, experimental systems and methods used in many studies. Here, indicate whether each material, system or method listed is relevant to your study. If you are not sure if a list item applies to your research, read the appropriate section before selecting a response.

## Materials & experimental systems

| n/a | Involved in the study |
|---|---|
| ☒ | ☐ Antibodies |
| ☒ | ☐ Eukaryotic cell lines |
| ☒ | ☐ Palaeontology and archaeology |
| ☒ | ☐ Animals and other organisms |
| ☒ | ☐ Clinical data |
| ☒ | ☐ Dual use research of concern |
| ☒ | ☐ Plants |

## Methods

| n/a | Involved in the study |
|---|---|
| ☒ | ☐ ChIP-seq |
| ☒ | ☐ Flow cytometry |
| ☐ | ☒ MRI-based neuroimaging |

## Magnetic resonance imaging

### Experimental design

| | |
|---|---|
| Design type | Event-related design |
| Design specifications | The design consisted of 300 decision trials per subject split into 2 runs, with a 5 minute break between runs, and each run taking 25 minutes. Between trials, there was a jittered inter-trial interval of between 2.5 and 8 seconds. At the onset of the option offers, participants were required to wait 2 seconds before they could indicate their response by button press, to maximally dissociate decision and motor response events. |
| Behavioral performance measures | Button press choices (between three offers) were recorded, alongside response times. Performance was quantified as the number of points won during the session, and all participants performed well above chance. |

## Acquisition

| | |
|---|---|
| Imaging type(s) | Functional and structural |
| Field strength | 3T |
| Sequence & imaging parameters | Siemens scanner with a multiband accelerated echoplanar imaging sequence with the following parameters: voxel resolution 2.4 x 2.4 x 2.4 mm3, repetition time=1230 ms, echo time=30ms, flip angle=60 degrees, field of view=240mm, multiband acceleration factor=3, PAT factor=2, encoding direction=PA. A tilt angle of 30 degrees was used to minimize signal drop out in the orbitofrontal cortex (Deichmann et al., 2003). |
| Area of acquisition | Whole brain |
| Diffusion MRI | ☐ Used  ☒ Not used |

## Preprocessing

| | |
|---|---|
| Preprocessing software | Data were pre-processed using FMRIB's Software Library ( FSL version 6.0.4), using the FEAT software tool (Woolrich et al. 2001). Gaussian spatial smoothing was applied with a full-width half-maximum of 5mm, and high pass temporal filtering was applied with a cut-off of 60s. |
| Normalization | Registration to standard space was performed using FLIRT (FMRIB's Linear Image Registration Tool) inside the FEAT software. Subject's T1-weighted structural image was used first to register the fMRI low resolution image. This was subsequently transformed to a standard T1-weighted image in MNI152 space. |
| Normalization template | Data were normalized to MNI152 space. |
| Noise and artifact removal | Functional data were motion corrected using rigid body registration to the central volume (Jenkinson et al., 2001, 2002). Cardiac and respiratory data were processed using FSL's Physiological Noise Modelling (PNM) tool to model the effects of physiological noise in the MRI data (Brooks et al. 2008). |
| Volume censoring | We detected and removed motion outliers using FEAT's fsl_motion_outliers tool. |

## Statistical modeling & inference

| | |
|---|---|
| Model type and settings | Univariate analysis methods were used. A general linear model (GLM) was used to model BOLD activity in pre-whitened data space using parametric event-related regressors. Seven regressors of interest were included in the main GLM, predicting BOLD activity at the onset of the decision period. These regressors included participant choices, and model value for the different options. |
| Effect(s) tested | Standard higher-order statistical tests were performed on the group data estimates. GLM parameter estimates were first estimated at the level of run (first level), then combined within individuals as Fixed Effects (second level), and finally combined across subjects using FMRIB's Local Analysis of Mixed Effects (FLAME1+2; third level; Woolrich et al. 2004). |

Specify type of analysis:  ☐ Whole brain  ☐ ROI-based  ☒ Both

| | |
|---|---|
| Anatomical location(s) | Regions of interest in vmPFC, ACC, and striatum were selected on the basis of activity peaks from orthogonal regressors identified from the whole-brain analysis. |

| | |
|---|---|
| Statistic type for inference | Cluster-wise analyses were performed using a cluster probability threshold of p=0.05. |

(See Eklund et al. 2016)

| | |
|---|---|
| Correction | Multiple comparisons were corrected for using a Z statistic threshold of 3.1. |

## Models & analysis

| n/a | Involved in the study |
|---|---|
| ☐ | ☒ Functional and/or effective connectivity |
| ☒ | ☐ Graph analysis |
| ☒ | ☐ Multivariate modeling or predictive analysis |

| | |
|---|---|
| Functional and/or effective connectivity | We used standard general linear models (GLM) to investigate how neural activity varied parametrically with model regressors, performing statistics primarily on the group data estimates. We probed these effects further on an individual level by extracting region-of-interest activity and investigating the relationship between traits and fMRI activity on the individual level (Spearman's correlation). |

