## [Peer Review File · Nature Human Behaviour]

Peer Review Information

Journal: Nature Human Behaviour

Manuscript Title: Goal commitment is supported by vmPFC through selective attention

Corresponding author name(s): Eleanor Holton

Reviewer Comments & Decisions:

Decision Letter, initial version:

18th September 2023

Dear Ms Holton,

Thank you once again for your manuscript, entitled "VmPFC supports persistence during goal pursuit through selective attention," and for your patience during the peer review process.

Your manuscript has now been evaluated by 2 reviewers, whose comments are included at the end of this letter. Although the reviewers find your work to be of interest, they also raise some important concerns. We are interested in the possibility of publishing your study in Nature Human Behaviour, but would like to consider your response to these concerns in the form of a revised manuscript that addresses all of the reviewers' points, before we make a decision on publication.

In sum, we invite you to revise your manuscript taking into account all reviewer and editor comments. We are committed to providing a fair and constructive peer-review process. Do not hesitate to contact us if there are specific requests from the reviewers that you believe are technically impossible or unlikely to yield a meaningful outcome.

We hope to receive your revised manuscript within two months. I would be grateful if you could contact us as soon as possible if you foresee difficulties with meeting this target resubmission date.

- Include a "Response to the editors and reviewers" document detailing, point-by-point, how you addressed each editor and referee comment. If no action was taken to address a point, you must provide a compelling argument. When formatting this document, please respond to each reviewer comment individually, including the full text of the reviewer comment verbatim followed by your response to the individual point. This response will be used by the editors to evaluate your revision and sent back to the reviewers along with the revised manuscript.
- Highlight all changes made to your manuscript or provide us with a version that tracks changes.

[REDACTED]

We look forward to seeing the revised manuscript and thank you for the opportunity to review your work. Please do not hesitate to contact me if you have any questions or would like to discuss these revisions further.

Sincerely,
Jamie

Dr Jamie Horder
Senior Editor
Nature Human Behaviour

REVIEWER COMMENTS:

Reviewer #1:

Remarks to the Author:

Holton et al. probe the mechanisms underpinning persistent goal pursuit using a lovely convergence of behavioral modeling, cross-task correlation, model-based fMRI, and human lesion-symptom mapping. This is one of the first papers I've seen to use model-based decomposition of choice behavior in a lesion group, which in itself is a novel and innovative contribution to the growing corpus of studies on vmPFC patients. The paper is timely and provides an important contribution. But, I am concerned with the ability of the empirical data to support the authors' interpretations on a few key points:

1. Potential confounding between "goal pursuit" and trial count / iteration#.

From my understanding of the main task, goal progress would often be directly proportional to trial count / iteration. As a result, I am concerned that the signal observed in vmPFC may be influenced by task-independent factors (e.g. increased reliance on model-free or habitual control as the task becomes more familiar). It would seem reasonable to include an iteration counter regressor to the

first-level GLM, orthogonalized to the goal pursuit regressor fit in Figures 3A-B.

2. Lack of consideration of choice confidence encoding in vmPFC.

Given the argument others have made about the role of confidence encoding in vmPFC during value-based choice (e.g. de Martino et al. 2012, *Nat Neuro*; Kepecs et al., 2008, *Nature*), I wonder if it also represents an alternative explanation for the current results. Specifically, pursuing the same goal on this paradigm would presumably be a relatively high-confidence choice. Despite frustrations, you know that adding more of the same will move you closer to a payout. In contrast, a late decision to abandon that goal and sample the alternative would presumably be a low confidence choice, uncertain about whether there will be enough trials remaining to fill their net when starting from scratch. Therefore, it seems possible that the “goal pursuit” regressor from their model would scale with choice confidence which may also scale with vmPFC BOLD in line with the existing vmPFC confidence literature. It would be good for the authors to include additional analyses (e.g. seeing whether vmPFC BOLD scales with trialwise RT as an indirect correlate of choice confidence) or additional thoughts in the discussion section to consider this possibility.

3. vmPFC ~ selective attention inference.

The authors seem to be making a stronger claim about the link their data can make between ventromedial prefrontal cortex and visuospatial selective attention than I believe is offered by the data. Specifically, there appear to be two possible ways attentional capture by the goal stimulus could arise in their (cleverly) interleaved spatial attention task: 1) through enhanced incentive salience of the goal itself (a “bottom-up” capture model), or 2) through enhanced memory-guided attentional orienting (a “top-down” anticipatory orienting model). The first argument would be in line with behavioral phenomena (summarized nicely in Pearson et al., 2022, *Nat Rev Psychol*), but I am unaware of imaging or lesion work linking vmPFC to this type of “bottomup” attentional capture, whereas the latter “top-down” account would be directly in line with studies on the functional role of vmPFC in memory-guided orienting of attention to goals (e.g. Günseli & Aly, 2020, *Elife*). The authors seem to argue that their results are in line with the bottom-up capture model, but the available correlation in Figure 2G does not seem to resolve these competing ideas about vmPFC’s functional role here. A consideration of the potential alternative explanation of memory-guided attention (either via additional analyses or thoughtful consideration of this issue in the discussion) would be a valuable addition to the paper.

Minor comments:

1. in several places in the manuscript a spearman rho is used whereas in others a pearson r is used without clarifying when / why parameteric vs. nonparametric tests were used on the Methods.
2. Fsl motion outliers was run on these data, but it is unclear what the scrubbing threshold was within subjects, or how much data had to remain after scrubbing to retain subjects in the second-level models. Given the potential impact of scrubbing timepoints on the BOLD autocorrelation—which can be particularly impactful when looking at trialwise parametric modeling of BOLD as in this paper—it would be good to know more about how this was implemented.

Reviewer #2:

Remarks to the Author:

Dear Editor and Authors,

Thank you for the opportunity to review this manuscript. I believe the described research is innovative in its design and thorough in its execution. The combination of the fMRI experiment with lesion patients especially provides additional support for the authors' narrative. While I'm not up-to-date on the various literature regarding the role of vmPFC, I can certainly understand the importance of the topic at hand. I am particularly impressed by how many of the comments or questions I was going to raise were addressed by the various analyses already included in the manuscript and the supplemental materials. I believe the manuscript has sufficient merit for publication.

I do have some questions to the authors so that I may better understand the manuscript.

1. With regards to the behavioral model comparisons, I find it surprising that the 'full task model' is the best fit. This is not to doubt the results of course. The main reservation is twofold for me. Firstly, while the lack of a formulaic model limits my understanding of the model, it seems like it is a rather computationally complex model with the addition of a very fine-tuned knowledge about the task-generative process. Hence, I'm unsure as to how viable of an assumption is it that participants have access to such computational capacity and task knowledge. Secondly, while the full-task model is quite a bit more complex than the other models, my read from the manuscript is that it had the same number of free parameters as the others and hence was not penalized much by the BIC metric. I worry as to whether the relative complexity of the model is sufficiently penalized.

Regarding these two points, which the authors can correct me if I'm mistaken, it would allay the concerns a bit if the authors could show that 1) the persistence bias is not specific to the chosen model and can be captured easily even with imperfect predictors or from model-free analyses, and 2) an out-of-sample metric such as choice prediction ROCs instead of an in-sample metric such as BIC. On the second point, I would think, given the relative simplicity of the linear model (despite the complexity of the calculations required for the regressors), the full model would likely perform the best. However, what might be beneficial is to see an interpretable metric of how much better the full model performs, as raw BIC values are a bit hard to fathom (e.g., does it predict 80% of choices as opposed to 70% by other models?).

2. I'm a little unsure on the term 'baseline vmPFC' attributed to the choice onset as opposed to the inter-trial period. This comes from reading a number of papers where value-related activities during decision-making were assessed with regressors time-locked onto the choice onset period. I may be tempted to interpret the results as vmPFC activity during early choice presentation. Wouldn't the inter-trial period vmPFC activity be more fitting of 'baseline'?

Minor comments:

In the logistic formula on page 28 under 'Model Validation Process', perhaps there is a negative sign missing in front of the exponent. As the formula currently stands, the probability of abandoning decreases as a function of $SV_{abandon}$.

Sincerely,
Sangil Arthur Lee

Author Rebuttal to Initial comments

Reviewer #1:

Remarks to the Author:

Holton et al. probe the mechanisms underpinning persistent goal pursuit using a lovely convergence of behavioral modeling, cross-task correlation, model-based fMRI, and human lesion-symptom mapping. This is one of the first papers I've seen to use model-based decomposition of choice behavior in a lesion group, which in itself is a novel and innovative contribution to the growing corpus of studies on vmPFC patients. The paper is timely and provides an important contribution. But, I am concerned with the ability of the empirical data to support the authors' interpretations on a few key points:

Thank you for your feedback and kind words. Below we will address your concerns point by point.

1. Potential confounding between “goal pursuit” and trial count / iteration#.

From my understanding of the main task, goal progress would often be directly proportional to trial count / iteration. As a result, I am concerned that the signal observed in vmPFC may be influenced by task-independent factors (e.g. increased reliance on model-free or habitual control as the task becomes more familiar). It would seem reasonable to include an iteration counter regressor to the first-level GLM, orthogonalized to the goal pursuit regressor fit in Figures 3A-B.

The reviewer rightly points out that it is important to control for task familiarity, as participants might rely more on habitual control or undergo other psychological changes, which may interact with vmPFC activity. First, we would like to begin by making the distinction between trial number within a block (which resets once a goal is complete), and trial number across the whole study. In our paradigm, trial number within a block is correlated with goal progress (but not task familiarity), and trial number over the study would reflect task familiarity (but does not correlate with goal progress).

We apologize for the confusion in the wording of the manuscript, as we realize this distinction between goal progress and overall study progress was not made clear. Below we show the correlation between goal progress and within-block trial number, and across-study trial number respectively.

Importantly, the plot does show that within-block trial number is correlated with progress (which is an inherent feature of the task that participants will have made the most progress late in the block). However, despite this correlation ($r=0.59$), trial number within a block and goal progress can still be differentiated because different goals progress at different rates, depending on factors such as the size of the goal, the option offers, and the participant's tendency to switch.

As suggested by the reviewer, below we explicitly control for (both interpretations of) trial number by including them as additional regressions in our main GLM. We present this analysis within our vmPFC region of interest, as this is the region relevant for our main results and identified by the reviewer as the potential area of confound. When controlling for both forms of trial number all within the same GLM, goal progress continues to impact activity at baseline, as shown below.

We thank the reviewer for bringing this confusion to our attention, and have included the following changes in the manuscript to reflect this:

1. In the first mention of goal progress, we now clarify the distinction between goal progress and study progress.

(c) Across individuals, persistence biases increased as a function of goal progress (i.e. the percentage of the net which had been completed). **Note that goal progress differs from overall progress in the study, since multiple goals (nets) are completed over the course of the study.**

2. In the supplementary materials, we include the plots shown above, with the following description

Supplementary Figure 10. Relationship between goal progress and trial number

(a, b) Goal progress (i.e. proportion of the goal completed) is correlated with trial number within a block (a; where a block corresponds to a single goal), but is not correlated with total trial number across the study (b).

(c, d, e) When controlling for the additional possible confounds of within-block trial number (d) and across-study trial number (e), vmPFC baseline activity tracks goal progress (c).

2. Lack of consideration of choice confidence encoding in vmPFC.

Given the argument others have made about the role of confidence encoding in vmPFC during value-based choice (e.g. de Martino et al. 2012, Nat Neuro; Kepecs et al., 2008, Nature), I wonder if it also represents an alternative explanation for the current results. Specifically, pursuing the same goal on this paradigm would presumably be a relatively high-confidence choice. Despite frustrations, you know that adding more of the same will move you closer to a payout. In contrast, a late decision to abandon that goal and sample the alternative would presumably be a low confidence choice, uncertain about whether there will be enough trials remaining to fill their net when starting from scratch. Therefore, it seems possible that the “goal pursuit” regressor from their model would scale with choice confidence which may also scale with vmPFC BOLD in line with the existing vmPFC confidence literature. It would be good for the authors to include additional analyses (e.g. seeing whether vmPFC BOLD scales with trialwise RT as an indirect correlate of choice confidence) or additional thoughts in the discussion section to consider this possibility.

We strongly agree with the reviewer that decision confidence is a key consideration for our paradigm. We therefore follow the reviewer’s suggestions in showing the relationship between response times and goal pursuit, and subsequently showing how this relates to neural activity. In addition, we have added detailed consideration of the potential role of confidence into the Discussion section of the manuscript.

Relationship between response times, goal pursuit, and decision type

Although response times (RTs) are slower on the first trial of a block compared to on subsequent trials (median first trial = 2.06 seconds, median post first trial = 1.67 seconds, Wilcoxon $z = 4.53$, $p < 0.001$) when excluding this initial trial we do not find a general relationship between goal progress and RTs on subsequent trials during goal pursuit. However, we do find a strong effect of the decision itself: participants are much slower to respond on abandonment trials compared to persist trials (mean RT on persist trial=1.52 seconds, $SD=0.28$; mean RT on abandonment

trial=2.59 seconds, $SD=0.62$, $t(29)=11.05$, $p<0.001$), reflecting the intuition that goal abandonment will generally be associated with lower choice confidence between the options. Below we show this by plotting mean RT for the different trial types (right), and against successive bins of goal progress (left).

Reaction times and neural analyses

We wholeheartedly agree with the reviewer that it is crucial to include RTs as an additional confound regressor in our neural analyses. For this reason, log reaction times were already included as a confound regressor in all of our fMRI analyses (both in the whole-brain analyses, and in our region-of-interest analyses). We apologize for not making this fact clear in the manuscript, and have added additional lines to the Results section shown at the end of this section to make sure this is clear to future readers.

From the whole-brain analysis, the areas where longer response times (as a proxy for lower confidence) predict more activity include ACC, insula, and dlPFC. The areas where shorter

response times (as a proxy for higher confidence) predict stronger activity include ventral striatum and frontal pole (area 9). In addition to the whole-brain analysis, we also show below the relationship between response time and neural activity within the three regions-of-interest discussed in the paper. Of critical relevance to the reviewer's concern, we do not find a significant relationship between the activity in our vmPFC region of interest and response times in our study. Since goal pursuit and response times are not very correlated in our task, and response times also do not drive vmPFC activity independently, vmPFC tracking of goal progress in our task cannot be explained simply by confidence as measured by response times.

One possible reason why we see response times driven more by the type of choice (initial trial, persistence, abandonment) rather than by goal progress in our paradigm, could relate to the general theory that goal pursuit attenuates evaluation between options by setting a default of persisting with the goal option. For this reason, trial-wise response times may not generally track the confidence associated with offers during goal pursuit unless a drastic shift in offer values prompts re-evaluation of the goal.

To address the significance of the reviewer's points regarding decision confidence within the manuscript, we have added the following extensions to the Results and Discussion sections of the manuscript:

Added to **Results** Section:

In contrast, ACC, presupplementary motor area (preSMA), bilateral dorsolateral prefrontal cortex (dlPFC), and bilateral insular, all showed the opposite profile: activity increased as the value of abandonment increased (value of best alternative–value of current goal; Fig.4a, orange), and activity was higher on trials where the participant chose to abandon the current goal (Supplementary fig.6b; See Supplementary tab.1 for activity peaks). **We included response times as an additional control regressor, previously argued to be a proxy for choice confidence (De Martino et al. 2013). We found that ACC activity was also higher when participants were slower to respond, but we found no relationship between response times and vmPFC activity (Supplementary fig.11c).**

Added to Discussion Section:

We have argued that activity in vmPFC relates to persistence with a goal. However previous studies have shown vmPFC activity is also related to decision confidence (De Martino et al. 2013, Kepecs et al. 2009). Since choices to persist with partially-completed goals tend to be associated with higher confidence than choices to start again with an alternative goal, it is important to consider whether this could be a potential confound. Using response times as a proxy for confidence, we did not find any relationship between vmPFC activity and response times, suggesting that vmPFC activity is not obviously related to simple measures of decision confidence in our task. One possible explanation for this disparity with previous findings could relate to the specific setting of incremental goal pursuit. If there is a strong default to persist with the current goal during goal pursuit, vmPFC activity may not track trial-wise variations in confidence associated with offers unless a drastic change prompts re-evaluation of the goal.

Added to Supplementary Materials

Supplementary Figure 11. Reaction times

(a) Relationship between reaction times and goal progress. Although RTs are slower on the first trial of a block compared to on subsequent trials (median first trial = 2.06 seconds, median post first trial = 1.67 seconds, Wilcoxon $z = 4.53$, $p < 0.001$) when excluding this initial trial we do not find a general relationship between goal progress and RTs on subsequent trials during goal pursuit (statistically tested using nested mixed effects models predicting reaction times with participant modelled as random effect. No significant model improvement is observed by adding goal progress as an additional fixed effect predicting reaction times: $X^2(1, N=30) = 0.41$, $p = 0.939$).

(b) Relationship between reaction times and trial type. Participants are much slower to respond on abandonment trials compared to persist trials (mean RT on persist trial=1.52 seconds, SD=0.28; mean RT on abandonment trial=2.59 seconds, SD=0.62, $t(29)=11.05$, $p<0.001$),

(c) Relationship between log reaction times and whole-brain activity, at the time of decision. The areas where longer RTs predict more activity include ACC, insula, and dlPFC. The areas where shorter RTs predict stronger activity include ventral striatum and frontal pole (area 9). We do not see activity related to RTs in vmPFC.

3. vmPFC ~ selective attention inference.

The authors seem to be making a stronger claim about the link their data can make between ventromedial prefrontal cortex and visuospatial selective attention than I believe is offered by the data. Specifically, there appear to be two possible ways attentional capture by the goal stimulus could arise in their (cleverly) interleaved spatial attention task: 1) through enhanced incentive salience of the goal itself (a “bottom-up” capture model), or 2) through enhanced memory-guided attentional orienting (a “top-down” anticipatory orienting model). The first argument would be in line with behavioral phenomena (summarized nicely in Pearson et al., 2022, Nat Rev Psychol), but I am unaware of imaging or lesion work linking vmPFC to this type of “bottom-up” attentional capture, whereas the latter “top-down” account would be directly in line with studies on the functional role of vmPFC in memory-guided orienting of attention to goals (e.g. Günseli & Aly, 2020, Elife). The authors seem to argue that their results are in line with the bottom-up capture model, but the available correlation in Figure 2G does not seem to resolve these competing ideas about vmPFC’s functional role here. A consideration of the potential alternative explanation of memory-guided attention (either via additional analyses or thoughtful consideration of this issue in the discussion) would be a valuable addition to the paper.

We thank the reviewer for raising a critical point about the alternative interpretations of the relationship between vmPFC and attention, which our previous version of the manuscript did not sufficiently discuss. We agree with the reviewer that our data cannot resolve which of these alternative attentional mechanisms causes the effect we see in Figure 2G. We have made two

changes to reflect this. First, within the results section we have removed any explicit claims that the attentional effect is either top-down or bottom-up. Second, we have added the following paragraph to the discussion bringing in the literature helpfully suggested by the reviewer, and providing a more thorough discussion of these possible explanations of the results.

There are limitations to how the attentional effects are interpreted in this study. The goal-directed attentional biases observed could stem from either memory-guided “top-down” orientation towards goal-relevant information (Nobre & Stokes, 2019, Günseli & Aly, 2020) or “bottom-up” attentional capture from the high incentive salience of goal-related stimuli (Anderson et al., 2011, Le Pelley et al., 2015, Cheng et al. 2021). While our task cannot definitively distinguish between these possibilities, one relevant consideration is how quickly attention shifts towards the new goal stimulus, rather than reflecting the reward history of stimuli across the study. This rapid attentional adaptation to new goals may be evidence in favour of goal-directed attentional orientation (although some studies have found goal-congruency to be stronger predictors of attentional capture than long-run value; Sepulveda et al. 2020, Frömer et al. 2019). The top-down attention interpretation is also consistent with previous evidence of vmPFC involvement in memory-guided attention (Small et al., 2003, Günseli & Aly, 2020).

Minor comments:

1. in several places in the manuscript a spearman rho is used whereas in others a pearson r is used without clarifying when / why parameteric vs. nonparametric tests were used on the Methods.

Many thanks for drawing out attention to this. We have now clarified that non-parametric Spearman’s correlations were used because persistence biases were not normally distributed. This is reflected in the Methods section:

- (1) We tested for a relationship between an individual’s goal-oriented attentional bias, and their persistence. Spearman’s correlation was used because as previously noted, persistence biases violated assumptions of normality.

(2) Spearman's correlation was used because both the neural activity betas and persistence bias distributions violated the assumption of normality.

2. Fsl motion outliers was run on these data, but it is unclear what the scrubbing threshold was within subjects, or how much data had to remain after scrubbing to retain subjects in the second-level models. Given the potential impact of scrubbing timepoints on the BOLD autocorrelation—which can be particularly impactful when looking at trialwise parametric modeling of BOLD as in this paper—it would be good to know more about how this was implemented.

We have extended the Methods section to reflect the reviewer's important question regarding the procedure for motion outlier detection. Note that we did not eliminate any subjects from further analysis due to motion, but instead controlled for timepoints within the GLM by adding a matrix of confound regressors to control for the motion outlier time-points. The median percentage of time-points scrubbed was 2.40%, while the maximum across all runs and all subjects was 9.10%. Below we show a plot of the distribution of proportion of timepoints scrubbed using this method, across all runs.

We have revised the methods section of the manuscript to include the following information regarding motion outlier detection procedure:

Motion outliers were detected using FEAT's `fsl_motion_outliers` tool. Metric values for detecting motion outliers were calculated for each time-point using the RMS intensity difference between each volume and the reference volume, and outliers were identified as volumes for which the metric value exceeded the 75th percentile + 1.5 times the InterQuartile Range. Note that no subjects or time-points were removed from analyses due to motion, but rather the effect of these outlier timepoints on the analysis was controlled for by including a confound matrix of outlier timepoints in the GLM. Across runs, the median percentage of time-points identified as outliers was 2.4% of volumes (max across all runs and all subjects=9.1%).

Reviewer #2:

Remarks to the Author:

Dear Editor and Authors,

Thank you for the opportunity to review this manuscript. I believe the described research is innovative in its design and thorough in its execution. The combination of the fMRI experiment with lesion patients especially provides additional support for the authors' narrative. While I'm not up-to-date on the various literature regarding the role of vmPFC, I can certainly understand the importance of the topic at hand. I am particularly impressed by how many of the comments or questions I was going to raise were addressed by the various analyses already included in the manuscript and the supplemental materials. I believe the manuscript has sufficient merit for publication.

I do have some questions to the authors so that I may better understand the manuscript.

Thank you for your thoughtful reviews of the manuscript. We hope that our responses below address the insightful questions that you had.

1. With regards to the behavioral model comparisons, I find it surprising that the 'full task model' is the best fit. This is not to doubt the results of course. The main reservation is twofold for me. Firstly, while the lack of a formulaic model limits my understanding of the model, it seems like it is a rather computationally complex model with the addition of a very fine-tuned knowledge about the task-generative process. Hence, I'm unsure as to how viable of an assumption is it that participants have access to such computational capacity and task knowledge. Secondly, while the full-task model is quite a bit more complex than the other models, my read from the manuscript is that it had the same number of free parameters as the others and hence was not penalized much by the BIC metric. I worry as to whether the relative complexity of the model is sufficiently penalized.

Regarding these two points, which the authors can correct me if I'm mistaken, it would allay the concerns a bit if the authors could show that 1) the persistence bias is not specific to the chosen model and can be captured easily even with imperfect predictors or from model-free analyses, and 2) an out-of-sample metric such as choice prediction ROCs instead of an in-sample metric such as BIC. On the second point, I would think, given the relative simplicity of the linear model (despite the complexity of the calculations required for the regressors), the full model would likely perform the best. However, what might be beneficial is to see an interpretable metric of how much better the full model performs, as raw BIC values are a bit hard to fathom (e.g., does it predict 80% of choices as opposed to 70% by other models?).

We strongly agree with all the points made by the reviewer, and thank them for the helpful suggestions. We have taken both suggestions on board, which are addressed in turn below.

1. Independence of the persistence bias metric from assumptions of the behavioral model

Throughout the paper, we compute an individual's *persistence bias* as their bias away from an approximately optimal model, which is a tree-search model which makes choices based on simulations of future offers using the generative model of the task. Given the significance of this persistence bias metric to the main results of the paper, the reviewer suggested the argument

would benefit from showing that the main results do not depend on full endorsement of the tree-search model (previously referred to as the ‘full-task model’) as the best determinant of behavior, or indeed the optimal model. We wholeheartedly agree with this intuition, and therefore below we show below that persistence bias is closely correlated with model free measures of persistence (i.e. total number of abandonment choices in the study). We also replicate our behavioral finding that attentional capture in the inter-trial task is related to an individual’s persistence, but here using the model-free metric rather than model-based metric used in the main text. These two analyses are presented below.

a) **The correlation between the model-based persistence bias and the model-free measure of persistence** (i.e. number of abandonment choices). In this analysis, we take an individual’s total number of abandonment choices as a model-free metric of their persistence (whereby the fewer the number of abandonment choices across the study, the greater an individual’s persistence). Note that this is an imperfect measure as it will also depend on differences between the schedules completed by each participant. Model-based persistence bias (deviation from the tree-search model) is highly correlated with model-free abandonment. Intuitively, this shows people who make more choices to abandon the current goal (model-free) are less biased towards persisting (model-based). It also shows that our model-based metric is capturing the variability in the raw data.

b) **The correlation between the model-free metric of persistence, and the extent to which the current goal stimulus is prioritized in attention** (from the spatial task). Here we show that the relationship between attention and decision-making does not depend on using the model-based metric of persistence in decision-making. In fact, we see a strong relationship between attention and the model-free metric (total number of abandonment choices). The plot below shows that individuals who make more abandonment choices show lower prioritization of the goal stimulus in attention, consistent with the finding that lower model-based persistence biases predict lower goal-oriented attention.

We have added both plots to the supplementary materials, with the following description.

Supplementary Figure 12. Model-free metrics of persistence.

Supplementary Figure 12. Model-free metrics of persistence.

In this analysis, we take an individual's total number of abandonment choices as a model-free metric of their persistence (whereby the fewer the number of abandonment choices across the study, the greater an individual's persistence). Note that this is an imperfect measure as it will also depend on differences between the schedules completed by each participant.

(a) Model-based persistence bias (deviation from the tree-search model) is highly correlated with model-free abandonment (Spearman's $r = -0.76$, $p < 0.001$, 95% CI = (-0.56, -0.88)). Intuitively, this shows people who make more choices to abandon the current goal (model-free) are less biased towards persisting (model-based).

(b) Correlation between the model-free metric of abandonment, and an individual's goal prioritization in the interleaved attention task. Individuals who make more abandonment choices show lower prioritization of the goal stimulus in attention (Spearman's $r = -0.54$, $p = 0.002$, 95% CI = (-0.22, -0.75)).

2. Use of an out-of-sample metric for model comparison

The reviewer suggested that using an out-of-sample metric would be a better approach for evaluating between models, since the models differ in their computational complexity yet do not differ in the number of free parameters fitted. We completely agree with the reviewer that out-of-sample metrics would be more appropriate than traditional BIC methods, and thank them for this pertinent suggestion. We have subsequently implemented a leave-one-out cross-validation as shown (below). With the leave-one-out analysis, we confirm that the full-task model (which we now call 'tree-search' model) is still the best fit to behavior, when measured using out-of-sample predictive accuracy.

We have replaced the BIC fits with the leave-one-out cross-validated results in the main manuscript, and have described the leave-one-out cross-validation process in the methods section:

Model fitting

Participant data was aggregated across the scanner session (300 trials) and post-scanner session (100 trials) before model fitting. In each case, the model value of switching was calculated as the model's value for the current goal subtracted from the model's value for the best alternative goal. To determine the best fitting normative model, we fit the following models to behaviour:

1. $SV_{abandon} = \beta_0 + \beta_1 * (\text{alternative value}_{offer-max} - \text{goal value}_{offer-max})$
2. $SV_{abandon} = \beta_0 + \beta_1 * (\text{alternative value}_{myopic} - \text{goal value}_{myopic})$
3. $SV_{abandon} = \beta_0 + \beta_1 * (\text{alternative value}_{prospective} - \text{goal value}_{prospective})$
4. $SV_{abandon} = \beta_0 + \beta_1 * (\text{alternative value}_{full-task} - \text{goal value}_{full-task})$

We fit these models in a mixed effects logistic regression analysis predicting abandonment choices, where intercept and slope were also modelled as random effects across participants.

We used a leave-one-out cross validation process to evaluate between models, since the models differed in their conceptual complexity but not in the number of fitted parameters. For each participant, we fit each of the mixed-effects model to the choices of all other participants ($n=29$). For the held-out participant, we then computed the predicted abandonment value for each trial, and transformed this into the predicted probability of switching using the softmax function:

$$P_{abandon} = \frac{1}{1 + e^{-SV_{abandon}}}$$

We took the absolute difference between the predicted probability of switching, and each held-out participant's true response, and subtracted from 1 to compute the model accuracy for each participant separately. This allowed us to evaluate both the overall cross-validation accuracy of each model, as well as the frequency of best-fitting models across participants (Supplementary Fig.3).

The results of this cross-validation procedure can be seen below, where the tree-search (previously referred to as 'full-task') model is the best predictor of overall behaviour (a), both when predicting abandonment trials (c) and when predicting persistence trials (d). We also computed the frequency with which each model was most accurate at predicting held-out

participants (b). We found that the tree-search model was the best predictor in 27/30 cases, while the myopic model was the best predictor in 3/30 cases.

2. I'm a little unsure on the term 'baseline vmPFC' attributed to the choice onset as opposed to the inter-trial period. This comes from reading a number of papers where value-related activities during decision-making were assessed with regressors time-locked onto the choice onset period. I may be tempted to interpret the results as vmPFC activity during early choice presentation. Wouldn't the inter-trial period vmPFC activity be more fitting of 'baseline'?

We thank the reviewer for bringing this confusion to our attention, and apologize that this analysis was not well described in the manuscript. The reviewer is concerned by the use of ‘baseline activity’ to describe vmPFC activity at the time of onset. We believe this reflects the fact that we did not adequately describe how our analysis procedure looking at baseline activity deviates from the standard procedure which looks at decision-related activity.

We believe the reviewer is describing the standard analysis procedure for identifying decision-related activity. In those cases, activity time-locked to the choice onset period is convolved by the haemodynamic response function to capture the neural response to the choice presentation. In contrast, we want to clarify that in our analysis, we do not convolve activity by the haemodynamic response function, but rather we take the raw BOLD activity at exactly a single time-point at the moment of choice presentation ($t=0$). Due to the nature of the haemodynamic delay which typically peaks three to five seconds after stimulus presentation (Martindale et al. 2003), this activity at $t=0$ cannot relate to the information presented at choice onset, but rather reflects the state of pre-decision activity.

This definition of ‘baseline’ has been used in previous literature investigating how pre-decision vmPFC activity affects choice behavior (Vinckier et al. 2018). However, we note that while some previous studies take activity at exactly $t=0$ of choice onset as we did (Vinckier et al. 2018), other studies take activity at a time-point slightly earlier, for example 2 seconds prior to choice onset in one case (Lopez-Persem et al. 2016). Therefore, we replicated our analysis using activity 2 seconds prior to the choice onset, and show below that the relationship between baseline vmPFC activity and behavior do not substantially change if we use an earlier time-point to define baseline.

Original analysis: vmPFC baseline is defined as activity at t=0 of choice onset

Control: vmPFC baseline is defined as activity -2 seconds before choice onset

Notably, when examining a time-point two seconds prior to the onset, we see a slightly stronger relationship between attentional goal-prioritization and vmPFC activity, but a slightly weaker relationship between decision-making biases and neural activity. While we do not want to over-interpret this observation, we note this is consistent with the ordering of these two behavioral tasks: specifically, the attention task is always performed immediately before the choice onset. Therefore, even though the attention task is performed in an entirely separate session outside the scanner, we find an even stronger relationship between neural activity and attention biases at a time-point in the fMRI session closer to when the spatial task *would* be performed.

Second, we want to draw attention to an analysis within the supplementary materials where we followed the non-baseline (traditional) procedure of convolving activity after choice onset by the haemodynamic response function. We note that when looking at the decision-related activity in this way, the relationship between neural activity and persistence bias is not significant. Therefore, we believe our finding is specific to *baseline* activity: baseline tracking of goal progress in vmPFC predicts individual differences in goal-prioritization in attention and decision-making, while decision-related activity does not seem to predict these differences.

We apologize again for the confusion about this analysis, and have made the following changes to the manuscript in order to reflect this:

1. In the main text, we explicitly clarify how our procedure differs from a traditional decision-related activity approach

We follow previous studies by defining baseline activity as the unconvolved neural activity at the onset of offers (Vinckier et al. 2018). Note that due to the nature of the haemodynamic delay, this analysis captures pre-decision activity rather than decision-related activity, which is standardly taken by convolving activity with the haemodynamic response function (see Supplementary fig.13 for correlations with earlier pre-decision activity).

2. In the supplementary materials, we have included the baseline activity time-locked to two seconds prior to choice onset, as an additional control

Supplementary Figure 13. Relationship between vmPFC goal-tracking and behavior at an earlier pre-decision time-point

In the main manuscript, baseline vmPFC was defined as unconvolved activity at the moment ($t=0$) of choice-onset (as in Vinckier et al. 2018.) Here we show the relationship between baseline vmPFC and behaviour is unaffected if baseline activity is defined as the unconvolved activity two seconds prior to choice-onset (as in Lopez-Persem et al. 2016). In other words, to address concerns that activity could be related to the onset of the choices, here we show that activity at a time-point two seconds earlier is predictive of behavioural biases, consistent with the interpretation that pre-decision activity is the critical predictor. In contrast, Supplementary fig.8c shows that decision-related activity (post choice onset), is not predictive of individual differences.

Minor comments:

In the logistic formula on page 28 under 'Model Validation Process', perhaps there is a negative sign missing in front of the exponent. As the formula currently stands, the probability of abandoning decreases as a function of SVabandon.

Apologies for the error and many thanks for noticing. We have corrected the formula in the manuscript to include the missing negative sign.

References

Anderson, B. A., Laurent, P. A., & Yantis, S. (2011). Value-driven attentional capture. *Proceedings of the National Academy of Sciences*, 108(25), 10367–10371. <https://doi.org/10.1073/pnas.1104047108>

Cheng, P. (Xin), Rich, A. N., & Le Pelley, M. E. (2021). Reward Rapidly Enhances Visual Perception. *Psychological Science*, 32(12), 1994–2004. <https://doi.org/10.1177/09567976211021843>

De Martino, B., Fleming, S. M., Garrett, N., & Dolan, R. J. (2013). Confidence in value-based choice. *Nature Neuroscience*, 16(1), 105–110. <https://doi.org/10.1038/nn.3279>

Frömer, R., Dean Wolf, C. K., & Shenhav, A. (2019). Goal congruency dominates reward value in accounting for behavioral and neural correlates of value-based decision-making. *Nature Communications*, 10(1), 4926.

Günseli, E., & Aly, M. (2020). Preparation for upcoming attentional states in the hippocampus and medial prefrontal cortex. *eLife*, 9, e53191. <https://doi.org/10.7554/eLife.53191>

Kepecs, A., Uchida, N., Zariwala, H. A., & Mainen, Z. F. (2008). Neural correlates, computation and behavioural impact of decision confidence. *Nature*, 455(7210), Article 7210.

<https://doi.org/10.1038/nature07200>

Le Pelley, M. E., Pearson, D., Griffiths, O., & Beesley, T. (2015). When goals conflict with values: Counterproductive attentional and oculomotor capture by reward-related stimuli. *Journal of Experimental Psychology. General*, 144(1), 158–171. <https://doi.org/10.1037/xge0000037>

Lopez-Persem, A., Domenech, P., & Pessiglione, M. (2016). How prior preferences determine decision-making frames and biases in the human brain. *ELife*, 5, e20317.

<https://doi.org/10.7554/eLife.20317>

Martindale, J., Mayhew, J., Berwick, J., Jones, M., Martin, C., Johnston, D., Redgrave, P., & Zheng, Y. (2003). The hemodynamic impulse response to a single neural event. *Journal of Cerebral Blood Flow and Metabolism: Official Journal of the International Society of Cerebral Blood Flow and Metabolism*, 23(5), 546–555.

<https://doi.org/10.1097/01.WCB.0000058871.46954.2B>

Nobre, A. C., & Stokes, M. G. (2019). Premembering Experience: A Hierarchy of Time-Scales for Proactive Attention. *Neuron*, 104(1), 132–146. <https://doi.org/10.1016/j.neuron.2019.08.030>

Sepulveda, P., Usher, M., Davies, N., Benson, A. A., Ortoleva, P., & De Martino, B. (2020). Visual attention modulates the integration of goal-relevant evidence and not value. *eLife*, 9, e60705. <https://doi.org/10.7554/eLife.60705>

Small, D. M., Gitelman, D. R., Gregory, M. D., Nobre, A. C., Parrish, T. B., & Mesulam, M.-M. (2003). The posterior cingulate and medial prefrontal cortex mediate the anticipatory allocation of spatial attention. *NeuroImage*, 18(3), 633–641.

[https://doi.org/10.1016/S1053-8119\(02\)00012-5](https://doi.org/10.1016/S1053-8119(02)00012-5)

Vinckier, F., Rigoux, L., Oudiette, D., & Pessiglione, M. (2018). Neuro-computational account of how mood fluctuations arise and affect decision making. *Nature Communications*, 9(1), Article 1. <https://doi.org/10.1038/s41467-018-03774-z>

Decision Letter, first revision:

17th January 2024

Dear Dr. Holton,

Thank you for your patience as we've prepared the guidelines for final submission of your Nature Human Behaviour manuscript, "VmPFC supports persistence during goal pursuit through selective attention" (NATHUMBEHAV-23082537A). Please carefully follow the step-by-step instructions provided in the attached file, and add a response in each row of the table to indicate the changes that you have made. Please also address the additional marked-up edits we have proposed within the reporting summary. Ensuring that each point is addressed will help to ensure that your revised manuscript can be swiftly handed over to our production team.

We would hope to receive your revised paper, with all of the requested files and forms within two-three weeks. Please get in contact with us if you anticipate delays.

Nature Human Behaviour offers a Transparent Peer Review option for new original research manuscripts submitted after December 1st, 2019. As part of this initiative, we encourage our authors to support increased transparency into the peer review process by agreeing to have the reviewer comments, author rebuttal letters, and editorial decision letters published as a Supplementary item. When you submit your

final files please clearly state in your cover letter whether or not you would like to participate in this initiative. Please note that failure to state your preference will result in delays in accepting your manuscript for publication.

In recognition of the time and expertise our reviewers provide to Nature Human Behaviour's editorial process, we would like to formally acknowledge their contribution to the external peer review of your manuscript entitled "VmPFC supports persistence during goal pursuit through selective attention". For those reviewers who give their assent, we will be publishing their names alongside the published article.

Cover suggestions

We welcome submissions of artwork for consideration for our cover. For more information, please see our https://www.nature.com/documents/Nature_covers_author_guide.pdf target="new"> guide for cover artwork.

ORCID

Non-corresponding authors do not have to link their ORCIDs but are encouraged to do so. Please note that it will not be possible to add/modify ORCIDs at proof. Thus, please let your co-authors know that if they wish to have their ORCID added to the paper they must follow the procedure described in the following link prior to acceptance:

Nature Human Behaviour has now transitioned to a unified Rights Collection system which will allow our Author Services team to quickly and easily collect the rights and permissions required to publish your work. Approximately 10 days after your paper is formally accepted, you will receive an email in providing you with a link to complete the grant of rights. If your paper is eligible for Open Access, our Author Services team will also be in touch regarding any additional information that may be required to arrange payment for your article.

Please note that *Nature Human Behaviour* is a Transformative Journal (TJ). Authors may publish their research with us through the traditional subscription access route or make their paper immediately open

access through payment of an article-processing charge (APC). Authors will not be required to make a final decision about access to their article until it has been accepted. Find out more about Transformative Journals

[REDACTED]

Best regards,
Alex McKay
Editorial Assistant
Nature Human Behaviour

On behalf of

Jamie

Dr Jamie Horder
Senior Editor
Nature Human Behaviour

Reviewer #1:

Remarks to the Author:

The authors have made comprehensive and thoughtful edits to the manuscript that have addressed each

of the concerns I raised during the initial review. I have no further comments and think this manuscript would be ready for acceptance in its current form.

Reviewer #2:

Remarks to the Author:

Thank you very much for your thorough explanation of your analyses and the additional results. Apologies for misunderstanding your initial analyses with regards to my second point; it seems clear now in hindsight. I have no reservations in recommending publication of this manuscript.

Final Decision Letter:

Dear Ms Holton,

We are pleased to inform you that your Article "Goal commitment is supported by vmPFC through selective attention", has now been accepted for publication in *Nature Human Behaviour*.

Please note that *Nature Human Behaviour* is a Transformative Journal (TJ). Authors may publish their research with us through the traditional subscription access route or make their paper immediately open access through payment of an article-processing charge (APC). Authors will not be required to make a final decision about access to their article until it has been accepted. Find out more about Transformative Journals

With best regards,

Jamie

Dr Jamie Horder
Senior Editor
Nature Human Behaviour